# HLStrans: Dataset for C-to-HLS Hardware Code Synthesis

## Abstract

High-Level Synthesis (HLS) enables hardware design from C/C++ kernels but requires extensive transformations, such as restructuring code, inserting pragmas, adapting data types, and repairing non-synthesizable constructs, to achieve efficient FPGA implementations. While large language models (LLMs) show promise in automating these transformations, progress has been limited by the absence of large-scale, well-structured datasets. Existing HLS datasets focus primarily on resource estimation, lack paired C and HLS examples with testbenches, and cover only a narrow set of optimizations. We introduce HLStrans, the first benchmark-scale dataset for LLM-driven C-to-HLS synthesis. HLStrans contains over 124K paired C and HLS programs for real-world applications, with full testbenches and synthesis-based annotations of latency and resource usage. The dataset systematically captures five categories of transformations and is enriched by an automated augmentation pipeline combining LLMs, Monte Carlo Tree Search (MCTS), and Design Space Exploration (DSE). We benchmark state-of-the-art LLMs on HLStrans, demonstrating that retrieval and fine-tuning significantly improve success rates and performance.

## 1 Introduction

Specialized computing systems, particularly FPGAs, are increasingly deployed to accelerate compute-intensive workloads in domains such as machine learning, signal processing, and data analytics. High-Level Synthesis (HLS) has emerged as a key methodology for bridging software and hardware, allowing engineers to describe functionality in C/C++ and automatically generate hardware-ready RTL. However, generating high-performance HLS code is far from a direct translation: it requires structural code refactoring, insertion of optimization pragmas, adaptation of data types, replacement of functions with hardware-friendly intrinsics, and strict compliance with HLS coding styles. Therefore, we define the **C-to-HLS transformation** task as follows: given a sequential C/C++ kernel, generate a synthesizable HLS implementation that achieves efficient hardware acceleration on an FPGA platform. This task exemplifies the challenges at the intersection of AI and EDA, demanding not only correctness but also hardware-aware optimization. The impact of this task is described in Appendix A.5.

Recent work has demonstrated the potential of large language models (LLMs) for HLS code generation. Early studies explored direct translation from C++ to synthesizable HLS code, while others focused on automating pragma insertion, repairing unsynthesizable constructs, or leveraging retrieval-augmented and chain-of-thought prompting to improve optimization quality (Collini et al., 2024; Xiong et al., 2024; Bhattacharyya et al., 2024; Xu et al., 2024; Prakriya et al., 2025). While promising, these approaches are constrained by the lack of comprehensive benchmarks: existing evaluations are conducted on small, fragmented collections of kernels, making it difficult to reproduce results or compare methods fairly. Without a unified, large-scale dataset, it remains challenging to systematically assess or advance LLMs on the C-to-HLS task.

Although several datasets for HLS exist, such as HLSsyn (Bai et al., 2023), HLSDataset (Wei et al., 2023), MLSBench (Goswami et al., 2022), and HLSfactory (Abi-Karam et al., 2024), they fall short for this purpose. Most are designed for resource estimation rather than code transformation, are limited in scale (typically a few hundred to a few thousand kernels), and rarely include paired examples of original C code, optimized HLS code, and testbenches. Moreover, they capture only

a narrow slice of transformation diversity, focusing mainly on pragma insertion and overlooking critical steps such as code restructuring, data type adaptation, and repair of unsupported C constructs. As a result, current datasets cannot serve as a benchmark foundation for training or evaluating LLMs on realistic C-to-HLS synthesis.

To address these gaps, we present HLStrans[1], the first benchmark-scale dataset explicitly designed for LLM-driven C-to-HLS transformation. HLStrans contains over 124,200 C and HLS pairs drawn from diverse real-world applications, covering domains such as linear algebra, machine learning, DSP, image processing, and cryptography. Each entry includes a triple: the original C kernel, an optimized HLS implementation, and a validation testbench, with annotations of latency and resource metrics obtained via synthesis. The dataset systematically captures five categories of transformations: code restructuring, pragma insertion, data type adaptation, function replacement, and HLS-compliant repair, ensuring broad coverage of hardware-oriented optimizations. To further enrich this corpus, we introduce an automated augmentation framework that combines LLMs, Monte Carlo Tree Search (MCTS), and Design Space Exploration (DSE) to generate diverse, synthesizable variants guided by synthesis feedback.

In summary, our contributions are threefold:

- We release HLStrans, the first large-scale dataset for C-to-HLS transformation, enabling LLM training and fair benchmarking;
- We propose a novel augmentation pipeline that produces diverse, high-quality HLS implementations;
- We provide extensive evaluations of open-source and closed-source LLMs, showing that retrieval and fine-tuning on HLStrans significantly boost synthesis success rates and performance. By positioning HLStrans as both a resource and a benchmark, we aim to catalyze progress in LLM-powered hardware design and accelerate the integration of AI into future EDA workflows.

## 2 BACKGROUND AND RELATED WORKS

**LLM aided C to HLS.** There is an increasing body of literature on applying LLMs to generate HLS design from original C code. Collini et al. (2024) evaluates the basic task of translating naive C++ into synthesizable HLS C++. Bhattacharyya et al. (2024) demonstrates that LLMs can automate HLS pragmas and optimizations to produce synthesizable, high-performance RTL from C on image-processing benchmarks. Xu et al. (2024) presents an LLM-driven HLS program-repair framework that transforms C/C++ into synthesizable HLS-C. Xiong et al. (2024) extends this approach with retrieval-augmented generation and chain-of-thought prompting to deliver optimized HLS implementations across nine applications. However, to date, no work has evaluated LLM's capabilities transforming C code to HLS codes on a large-scale dataset.

**HLS code dataset.** HLSsyn (Bai et al., 2023) focuses on incorporating a diverse set of optimization pragmas but contains only 42 kernels for training and evaluating design-quality prediction models. HLSDataset (Wei et al., 2023), which aggregates 34 data sources into roughly 18K samples, targets power, resource, and timing estimation. MLSBench (Goswami et al., 2022) is an open-source corpus produced with the Xilinx Vivado HLS flow; it covers 17 C/C++ and 13 SystemC benchmarks, but provides only HLS log files and reports. DB4HLS (Ferretti et al., 2021) introduced a database of more than 100,000 HLS design points generated from MachSuite via exhaustive design-space exploration. Likewise, Dai et al. (2018) released about 1,300 designs created from benchmarks. Despite these valuable resources, they suffer from three key limitations when used to evaluate LLMs' ability to translate C code into HLS:

First, prior HLS datasets have primarily targeted **quality-of-results (QoR) estimation rather than C-to-HLS code generation**, and the underlying program sources are limited. Though varying tool configurations can yield many synthesized samples, the scarcity of distinct source programs prevents an LLM from learning diverse program structures needed for C-to-HLS tasks. Moreover, the selected programs are typically short, making them inadequate for fully assessing LLMs' capability.

Second, Existing datasets **inadequately capture comprehensive C-to-HLS transformations**, focusing largely on pragma insertion. Generating high-performance HLS code from standard C/C++

---

[1] https://anonymous.4open.science/r/HLStrans-B578/

for FPGAs requires a series of systematic transformations to expose parallelism, optimize data movement, and conform to HLS-friendly coding styles. While the detailed transformations are in Appendix A.1, these transformations fall into five broad categories, shown in Figure 1.

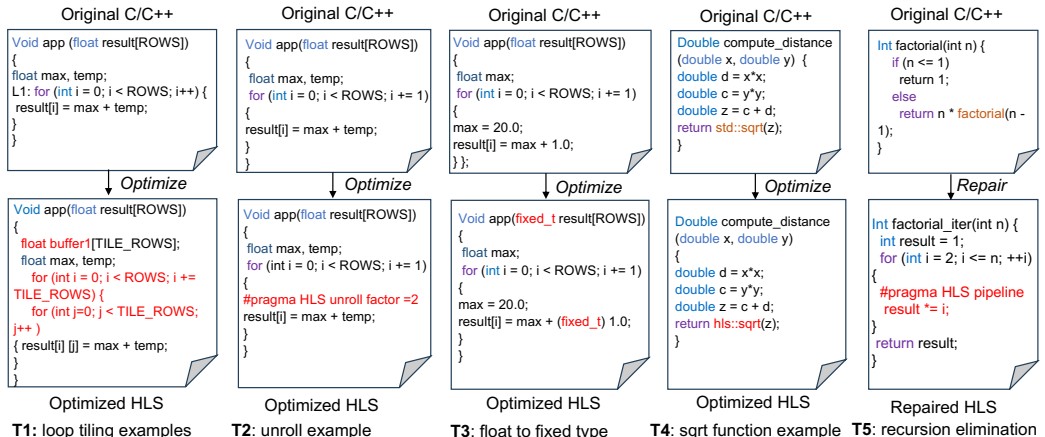

Figure 1: C/C++ to HLS code transformation examples. T1: Apply loop tiling and local buffering to improve data locality. T2: Unroll inner loops to increase parallelism and throughput. T3: Convert floating-point to fixed-point types to reduce resource use and latency. T4: Replace standard math calls with HLS intrinsics (e.g. hls::sqrt) for synthesizable implementations. T5: Eliminate recursion by refactoring to iterative code so the design can be synthesized.

*T1: Code Restructuring.* Refactor algorithms to expose pipelining and dataflow, apply loop tiling, memory coalescing, ping-pong buffering, and reorganize control logic for parallel or streaming execution. *T2: Directive (Pragma) Insertion.* Place HLS pragmas to guide the tool scheduler, such as data flow, pipeline, loop partition, and interface specifications, to fine-tune performance and resource usage. *T3: Data-Type Adaptation.* Replace generic C types with HLS-specific arbitrary-precision types: convert floating point to fixed point (*ap_fixed*) for resource optimization, standard integers to bit-accurate (*ap_int/ap_uint*), and customize bit widths to match application precision requirements. *T4: Transformation of Functions.* Transform standard C functions into HLS-optimized kernels or intrinsics (such as converting the *std::sqrt* function to the *hls::sqrt* function) to better leverage FPGA fabric and specialized accelerators. *T5: HLS-Compliant Coding Style.* Eliminate unsupported C constructs such as dynamic memory allocation (*malloc/free*), recursion, and certain pointer arithmetic patterns; restructure code to use static arrays, simple loops, and explicit hand-shaking for communication.

Third, they are not organized as **paired C-and-HLS examples and omit the corresponding testbenches** needed for LLM-based HLS code optimization, which are not ready for LLM to verify its output.

Compared with previous works, Table 1 concludes that our dataset has more kinds of sources and supports more transformations, making it ready for LLM code generation.

## 3 HLSTRANS DATASETS CONSTRUCTION

Open-source HLS datasets are scarce and poorly structured, which limits their usefulness for training LLMs. We propose an automated pipeline to generate high-quality HLS datasets from existing resources. The pipeline has three stages: (1) collect high-quality human optimized open-source HLS examples; (2) perform targeted data augmentation on human optimized kernels to produce many viable candidates; (3) select the efficient HLS implementations from those candidates. Figure 2 shows our dataset construction process.

### 3.1 DATASET COLLECTION

Firstly, we harvest code from GitHub, selecting repositories with at least ten stars. However, manually optimized codebases often exhibit inconsistent formatting and sparse documentation, which

Table 1: Comparison of HLS datasets. *QoR*: quality of result prediction. *Transformation*: C to HLS transformations mentioned in Figure 1. ✓: included. ✗: not included

| Attributes | Dai | MLSBench | DB4HLS | HLSdataset | HLSsyn | HLStrans |
|---|---|---|---|---|---|---|
| Samples | 1,300 | 6,000 | 124,106 | 18,876 | 42,000 | **124,200** |
| Programs | 65 | 30 | 19 | 34 | 42 | **309** |
| Purpose | QoR | QoR | QoR | QoR | QoR | **Code generation** |
| Transformations | T2 | T2 | T2 | T1,T2 | T2 | **T1, T2, T3, T4, T5** |
| Testbench | No | No | No | No | No | **Yes** |
| **Programs** | | | | | | |
| CHStone(Hara et al., 2008) | ✓ | ✓ | ✗ | ✓ | ✗ | ✓ |
| Polybench(Pouchet & Yuki, 2012) | ✗ | ✗ | ✗ | ✓ | ✓ | ✓ |
| Rodinia(Che et al., 2009) | ✗ | ✗ | ✗ | ✗ | ✗ | ✓ |
| Machsuite(Reagen et al., 2014) | ✓ | ✓ | ✓ | ✓ | ✓ | ✓ |
| Rosetta(Zhou et al., 2018) | ✗ | ✗ | ✗ | ✓ | ✗ | ✓ |
| C2HLS(Collini et al., 2024) | ✗ | ✗ | ✗ | ✗ | ✗ | ✓ |
| PP4FPGA(Kastner et al., 2018) | ✗ | ✗ | ✗ | ✗ | ✗ | ✓ |
| Forgebench(Wanna et al., 2025) | ✗ | ✗ | ✗ | ✗ | ✗ | ✓ |
| HLSfactory(Abi-Karam et al., 2024) | ✗ | ✗ | ✗ | ✗ | ✗ | ✓ |
| Others (GitHub) | ✗ | ✗ | ✗ | ✗ | ✗ | ✓ |

hinders LLM-driven code generation. Public kernels also frequently depend on unexpanded macros and bundle extraneous utility functions that obscure the core algorithm. To make C to HLS tasks readily consumable by LLMs, we package each design with the following files:

- Single original file $x$ that is a slow original C/C++ codes.
- Single optimized HLS file $y$ that implements the kernel, including a top function and, if necessary, any sub-functions and specialized data types. The file must be synthesizable and not exceed the resources of the platform.
- Self-contained C++ testbench $tb$ includes all test cases and validation logic necessary to verify the kernel's outputs against expected results. We manually write all the testbenches and adjust the optimized HLS code to ensure it passes all tests. The coverage of testbenches are described in Appendix A.7.

Therefore, we construct triples $(x, y, tb)$. If the original file $x$ is synthesisable, the execution cycles from synthesis reports of $y$ must be less than $x$. If the original file $x$ is not synthesisable, $y$ should be synthesisable.

## 3.2 DATASET AUGMENTATION

Relying solely on collected repositories is insufficient because high-quality hardware codes are far scarcer than general software. To generate richer, more useful examples, we designed an automated dataset-augmentation framework that synthesizes additional C to HLS variants.

We formulate the C to HLS translation as a combinatorial search problem: selecting appropriate combinations of code transformations to meet performance and resource targets. Our approach proceeds in two stages. First, an LLM agent guided by Monte Carlo Tree Search (MCTS) proposes and explores semantic-preserving code transformations that expose parallelism and produce HLS-friendly structure. Second, for each candidate design we apply automated design-space-exploration

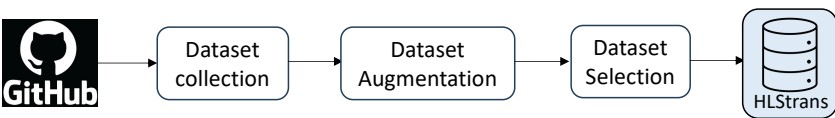

Figure 2: HLStrans Dataset Construction Process.

(DSE) tools to tune pragmas and low-level implementation choices. Both stages are evaluated in the loop using EDA feedback (performance, resource utilization, and compilation outcomes), enabling MCTS and DSE to efficiently navigate the large, combinatorial action space (see Figure 3).

First, MCTS performs structured exploration by balancing the exploitation of high-reward actions with the exploration of uncertain or less-visited regions of the search space. The optimization policy is generated by the retrieval augmentation generation (RAG) module. The search is guided to choose the suitable policy by both the verification pipeline and a reward model. The reward model incorporates detailed feedback from the HLS toolchain, including synthesis success or failure, compile warnings, and performance metrics such as resource usage, latency, and throughput. This heuristic-driven strategy enables the agent to iteratively refine transformation sequences and produce more high-quality, synthesizable HLS designs. Second, HLS directive design space exploration using genetic algorithms (Ferikoglou et al., 2023) is adopted. It inserts pipeline, unroll, and partition pragmas to produce more effective data samples. Through iterative refinement, the framework converges toward optimized and synthesizable HLS code.

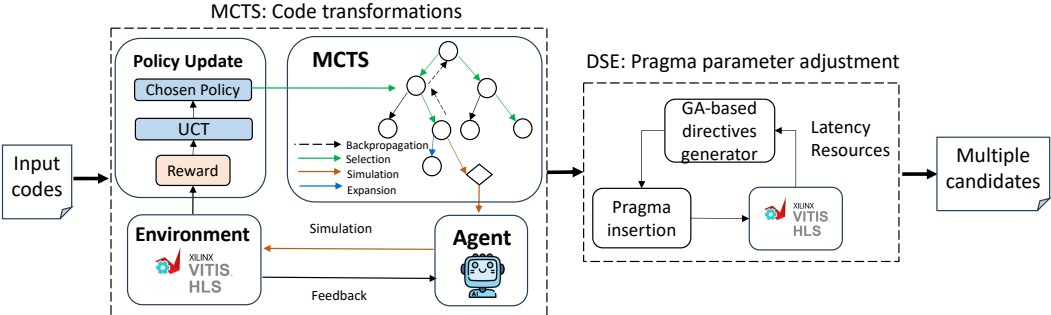

Figure 3: HLStrans Dataset Augmentation Framework.

### 3.2.1 MONTE CARLO TREE SEARCH (MCTS)

We formulate HLS optimization as an MCTS problem. The *environment* is the Vitis HLS toolchain, which provides synthesis, resource, and performance feedback. The *agent* is an LLM that applies code transformations. *Actions* include (i) RAG-based retrieval of known optimization policies and (ii) ReAct-based reasoning over compiler warnings. The *state* is the current HLS code, and the *reward* follows rule-based shaping: $-2$ for verification failure, $-1$ for synthesis/resource failure, $0$ if worse, $1$ if improved, and $2$ if improved with timing met. In our cases, the MCTS agent begins at the initial state $S_0$ (the root node), which is the naive HLS code. From a state $S_t$, the agent applies a optimization policy $\pi$, i.e., an action $a_t \in \mathcal{A}$, transitioning to the subsequent state $S_{t+1}$. This new state optimizes the existing code sequence by applying the new optimizations. Upon reaching a terminal state $S_T$, the agent receives a deferred reward $R(S_T)$. $N(S_t)$, the total number of times $S_t$ has been visited.

**Selection:** We employ the upper confidence bounds for trees (UCT) algorithm (Gelly & Wang, 2006) to choose nodes. The UCT formula includes the average reward for the current state, which encourages the path that can bring high reward, while $U$ term measures the associated uncertainty, which encourages the exploration of new paths. This approach effectively balances the trade-off between exploration and exploitation.

**Expansion, simulation and backpropagation:** Expansion is to explore the unchosen action. We leverage LLM to determine the next action from the unexplored. The decision process is driven by program analysis in conjunction with the history of adopted optimizations, enabling LLM to accurately assess and select the most promising action. After the analysis of LLM for state $s_t$ at time steps $t$, the next action $a_{t+1}$ will decided by $a_{t+1} = llm(s_t)$. Simulation employs the agent to apply transformations and evaluates them via HLS synthesis measuring estimated latency and resource usage to compute the reward $R(s_t, a_t)$; During backpropagation, these rewards are propagated up the search tree to update node values, refining the agent's estimates and guiding future action selection. Once we no longer observe significant improvements, the search process is halted, and the best-performing rewritten design is selected. The detailed description of MCTS framework is described in Appendix A.2.

### 3.2.2 DESIGN SPACE EXPLORATION

The tool implements an automated HLS design-space explorer that uses a genetic-algorithm optimizer to discover effective directive combinations, specifically loop pipelining, loop unrolling, and array partitioning, that maximize performance and resource utilization. To traverse the solution space, we utilize the NSGAII algorithm (Deb et al., 2002) implemented in PyMOO library (Blank & Deb, 2020), known for its ability to bypass local optimal and quickly converge to efficient solutions. The detail DSE implementation is introduced in Appendix A.2.2.

## 3.3 DATASET SELECTIONS

After generating multiple dataset candidates, we select the efficient samples. If the input codes can not be synthesized, we choose the candidates which can be synthesized. If the input codes can be synthesized, we choose the candidates whose latency is less than input codes. To give the model clearer guidance, we borrow ideas from Shypula et al. (2023) and attach a "performance tag" and "resource tag" to each solution during training. Each tag reflects how close that program comes to the best attainable performance with resources minimized, using a scale from 0 to 10, respectively.

## 3.4 HLSTRANS STATISTICS

Overall, we leverage DeepSeek-R1 to generate high-quality synthetic code examples, the AMD Vitis HLS EDA tool, and DSE tools to validate, annotate, and collect performance/resource metrics within our framework, yielding an effective HLS code dataset. Our dataset has the following merits:

**Diverse Application Coverage.** Table 1 shows that HLStrans provides the largest number of HLS kernels and the longest average lines of code, incorporating commonly used HLS benchmarks as well as real-world examples. Our curated corpus spans diverse application domains and covers all six transformation categories listed in Appendix A.1. Figure 4a visualizes the program distribution across these five tasks, and the source kernels themselves fall into seven distinct application categories. This rich, well-balanced dataset offers broad coverage of real-world HLS patterns required to train and evaluate LLMs' hardware-synthesis capabilities.

**Diverse types of transformations.** To evaluate the LLM's ability to assess different C/C++ to HLS transformations, every transformation shown in Figure 1 must be supported. Each dataset sample may correspond to one or more types of transformations. Figure 4b illustrates how the number of samples for each transformation increases after data augmentation.

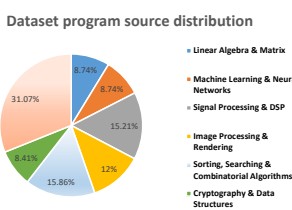
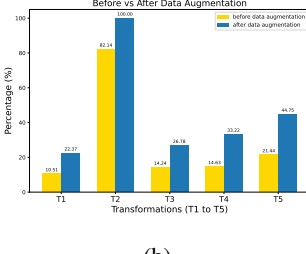
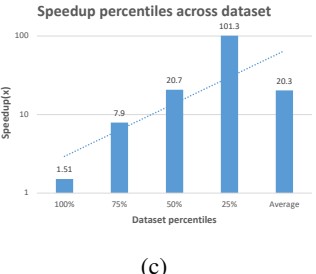

(a)  (b)  (c)

Figure 4: (a) Program source distribution. (b) Percentage of different transformations. (c) Speedup percentiles across dataset.

**High quality of dataset samples.** To evaluate the quality of the dataset, we measured the execution-cycle ratio between the original and target codes using reports from Vitis HLS. The speedup is ratio between the latency of the original design and the generated design, as reported by the synthesis tool. We then computed the percentile distribution of these speedup values across all pairs. As shown in the Figure 4c, 100% of the pairs, the target code is $\geq 1.5\times$ faster, and for 25% of the pairs, it achieves a speedup of $\geq 50.3\times$. Different samples are annotated with performance and resource usage tags, allowing the LLM to understand the detailed effects of C-to-HLS transformations. This enables the LLM to generate code that achieves higher performance while consuming fewer resources. The detailed information on dataset generation is in Appendix A.2. The datasets

are released under the MG0-2.0 Non-Commercial (NC) license (Duan et al., 2024).[2] These licenses permit both academic and commercial reuse provided that attribution is given. The dataset release also includes provenance metadata and third-party license notices.

# 4 Experimental Evaluation of LLMs on Our Dataset

To evaluate LLM performance on our dataset and assess the dataset's impact on model capability, we explore different prompting strategies and fine-tune smaller models using supervised fine-tuning (SFT) (Ouyang et al., 2022).

## 4.1 Prompting Methods

**Zero-shot Prompting:** We craft concise, HLS-specific prompts that instruct the model to perform code optimization or transformation from its pretrained knowledge, without any additional fine-tuning or example demonstrations (Liu et al., 2021) (Wei et al., 2021). **Chain-of-Thought Prompting:** Building on the chain-of-thought approach of Wei et al. (2022), our prompts first guide the model through a transformation reasoning phase before asking it to emit the refined code. **Retrieval-Based Prompting:** Recent studies (Shrivastava et al., 2023) (Shypula et al., 2023) have shown that retrieval-based techniques can substantially boost code generation quality in large language models. In our approach, we first encode each program using CodeBERT (Zhou et al., 2023) to produce rich, semantically informed embeddings. We then index these vectors with FAISS (Johnson et al., 2019) (Facebook AI Similarity Search) and perform a K-nearest-neighbors lookup to retrieve the top K most similar code snippets from our training corpus. Finally, we supply these retrieved examples alongside the original code as additional context to the LLM, guiding it to produce more accurate and effective edits. In our experiments, we set K to 1. The detailed prompt information is in Appendix A.3. The following experiments includes results of Vitis_HLS tools. Results of other HLS tools are shown in Appendix A.6.

## 4.2 Experiment setting

To evaluate LLMs on our dataset, we have the following evaluation setting. **Task setup:** Given a C/C++ kernel, the model must generate an optimized HLS implementation. Success requires not only functional correctness but also synthesizability under FPGA toolchains. **Models:** We benchmark both closed-source (GPT-5 (Wang et al., 2025), DeepSeek-R1 (Chua & Evans, 2025), Grok 4 (xAI, 2025), Gemini 2.5 Pro (Comanici et al., 2025)) and open-source (Qwen 2.5 Coder (Hui et al., 2024)) models, under different prompting strategies (zero-shot, chain-of-thought, retrieval) and fine-tuning. **Dataset split:** Following standard machine-learning protocol, we reserve 270 applications for training and validation and hold out 39 applications for evaluation. The held-out set includes both unsynthesizable designs that require repair and synthesizable designs that require optimization. Crucially, these 39 held-out applications were excluded from the LLM-based data-augmentation pipeline to prevent any risk of data leakage into the evaluation. **Infrastructure:** All synthesis is conducted with the Xilinx Vitis HLS toolchain targeting a datacenter FPGA (Alveo U55C). Training was conducted using 2 NVIDIA H100 GPUs, each with 80 GB of memory. The computing environment was configured with CUDA 12.2 and cuDNN 9.1 to ensure optimal deep learning performance. **Metrics:** Unlike conventional code generation benchmarks that stop at functional correctness, the C-to-HLS task requires models to satisfy both software and hardware constraints. We therefore report four complementary metrics:

- Functional Accuracy: The share of test programs that preserve the original functionality testbench.
- Synthesis Accuracy: Percentage of programs that compile successfully into FPGA-ready hardware.
- Speedup (Latency reduction): Ratio between the latency of the original design and the generated design, as reported by the synthesis tool.
- Optimization Rate (%OPT): Fraction of generated programs that both pass correctness checks and achieve speedup $> 1\times$.

---

[2] https://www.modelgo.li/.

Previous works (Li et al., 2022) show that generating multiple program candidates per input and selecting the optimal one improves code synthesis performance. We generate $k$ program variants for each input, then select the fastest one that successfully passes all test cases; we refer to this sampling-and-selection strategy as $Best@k$.

### 4.3 EXPERIMENT RESULTS

#### 4.3.1 EVALUATION OF DATA AUGMENTATION FRAMEWORK.

The MCTS component of our framework can produce variable iteration lengths. To quantify this behavior, we evaluated both runtime and achieved speedup on the PolyBench suite (Pouchet & Yuki, 2012) while sweeping the number of rollouts. Figure 5a reports these results and indicates that 32 rollouts is the "sweet spot"; consequently, we set the rollout count to 32 for subsequent experiments. We also compared our framework against state-of-the-art approaches on the Rodinia benchmarks by measuring the runtime of the optimized programs on real FPGA cards. Figure 5b shows the runtime comparison for five Rodinia benchmarks (Che et al., 2009). With the same base model, our framework attains more than $5\times$ average speedups than Xiong et al. (2024). We additionally observed that DeepSeek-R1 produces even better results; therefore, we selected DeepSeek-R1 as the generator for new data used in later experiments. These findings motivated both our choice of rollout parameter and our selection of the data-generation model. The detailed experiments comparison results are in Appendix A.2. The detailed experiments analysis are in Appendix A.9

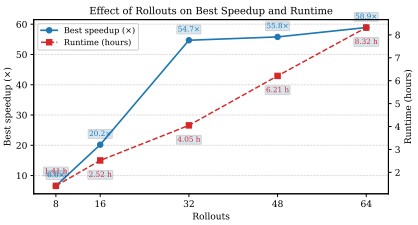

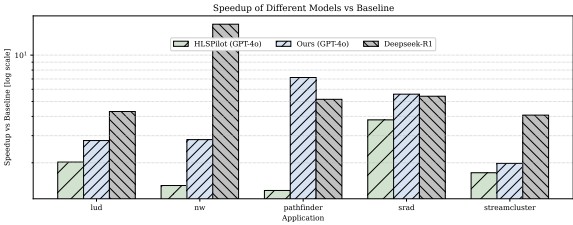

(a) The effect of MCTS rollout setting  (b) Comparison with sota framework

Figure 5: Evaluation of our dataset augmentation framework (a) MCTS rollout setting. (b) Rollout comparison.

#### 4.3.2 RESULTS OF FINE-TUNING MODELS

Both the Qwen2.5 Coder 3B and 7B fine-tuned models show consistent gains in optimization quality, latency reduction, and synthesis success rate in Table 2. They generate HLS code that not only executes faster but also synthesizes more reliably, even though the function-correct rate has slightly dropped. These results demonstrate that training on our curated dataset significantly boosts an LLM's ability to produce correct, high-performance HLS implementations directly from C sources.

Table 2: Fine-Tuning results comparison. *Transformation*: the T1–T5 transformation in Figure 1 applied to examples that are functionally and synthesis correct.

| Method | Model | Speedup | | | | Transformation | | | | | Functional Accuracy | Synthesis Accuracy |
|--------|-------|-----|-----|-----|-----|----|----|----|----|----|---------|----------|
| | | Opt | Min | Avg | Max | T1 | T2 | T3 | T4 | T5 | | |
| Pretrain | Qwen coder 7B | 2.6% | 0.27× | 1.03× | 3.6× | 0 | 5.1% | 0 | 0 | 2.6% | 12.8% | 10.3% |
| | Qwen coder 3B | 0% | 0.38× | 0.97× | 1× | 0 | 2.6% | 0 | 0 | 2.6% | 7.7% | 10.3% |
| SFT | Qwen coder 7B | **15.4%** | **0.6×** | **4.2×** | **21.8×** | 5.1% | **20.5%** | **5.1%** | **5.1%** | 2.6% | **20.5%** | **28.2%** |
| | Qwen coder 3B | 10.3% | 0.4× | 3.7× | 17.2× | 5.1% | 17.9% | 2.6% | 5.1% | 2.6% | 17.9% | 20.5% |

Efficient HLS kernels require a mix of C to HLS transformations. We measure how our dataset improves LLM C to HLS optimization: Table 2 demonstrates that fine-tuning on our corpus raises success rates across transformation types.

#### 4.3.3 RESULTS OF PRETRAINED MODELS

Table 3 reports the *Best@1* and *Best@5* accuracies for different prompts and models. To evaluate the utility of our dataset, we constructed retrieval databases from the HLSdataset (Wei et al., 2023) and

from our training data, and applied retrieval-based prompting using these two databases to measure its effect. Overall, incorporating our dataset into retrieval improved pretrained models performance compared with other prompt methods.

Table 3: Best@1 and Best@5 results for various methods and models.

| Method | Model | Best@1 | | | | | | Best@5 | | | | | |
|---|---|---|---|---|---|---|---|---|---|---|---|---|---|
| | | Speedup | | | | Functional Accuracy | Synthesis Accuracy | Speedup | | | | Functional Accuracy | Synthesis Accuracy |
| | | Opt | Min | Avg | Max | | | Opt | Min | Avg | Max | | |
| Zero-shot | Deepseek-R1 | 20.5% | 0.17× | 1.82× | 16.03× | 43.6% | 38.5% | 23.1% | 0.19× | 1.97× | 16.15× | 46.2% | 51.3% |
| | GPT-5 | 20.5% | 0.04× | 14.32× | 506.07× | 48.7% | 48.7% | 23.1% | 0.34× | 14.35× | 506.07× | 53.8% | 61.5% |
| | Grok-4 | 20.5% | 0.48× | 2.35× | 46.51× | 43.6% | 43.6% | 33.3% | 0.50× | 2.46× | 46.84× | 56.4% | 53.8% |
| | Gemini-2.5-pro | 25.6% | 0.98× | 2.74× | 35.57× | 41.0% | 41.0% | 30.8% | 1.21× | 2.89× | 36.01× | 46.2% | 51.3% |
| | Qwen coder 32B | 10.3% | 0.28× | 1.10× | 3.71× | 56.4% | 53.8% | 17.9% | 0.43× | 1.22× | 4.09× | 59.0% | 56.4% |
| COT | Deepseek-R1 | 25.6% | 0.21× | 2.1× | 19.01× | 48.7% | 46.2% | 28.2% | 0.34× | 2.18× | 19.07× | 51.3% | 53.8% |
| | GPT-5 | 25.6% | 0.19× | 17.1× | 425.07× | 53.8% | 53.8% | 38.5% | 0.41× | 17.12× | 437.07× | 56.4% | 66.7% |
| | Grok-4 | 20.5% | 0.53× | 2.56× | 49.77× | 48.7% | 51.3% | 33.3% | 0.82× | 2.92× | 50.12× | 61.5% | 53.8% |
| | Gemini-2.5-pro | 30.8% | 0.98× | 2.98× | 37.57× | 46.2% | 46.2% | 41.0% | 1.28× | 3.39× | 37.58× | 51.3% | 56.4% |
| | Qwen coder 32B | 15.4% | 0.37× | 1.910× | 5.87× | 61.5% | 59.0% | 20.5% | 0.52× | 2.03× | 6.25× | 71.8% | 66.7% |
| Retrieval Prompt (HLSdataset) | Deepseek-R1 | 20.5% | 0.47× | 30.10× | **953.30×** | 33.3% | 28.2% | 23.1% | **0.60×** | 30.28× | 954.41× | 35.9% | 35.9% |
| | GPT-5 | 18.0% | 0.01× | 1.96× | 31.60× | 33.3% | 28.2% | 30.8% | 0.23× | 2.04× | 33.86× | 35.9% | 41.0% |
| | Grok-4 | 12.8% | 0.07× | 1.59× | 19.19× | 33.3% | 25.6% | 25.6% | 0.36× | 2.32× | 26.23× | 46.2% | 28.2% |
| | Gemini-2.5-pro | 18.0% | 0.02× | 5.57× | 137.19× | 33.3% | 28.2% | 28.2% | 0.32× | 6.39× | 137.35× | 38.5% | 38.5% |
| | Qwen coder 32B | 10.3% | 0.33× | 2.67× | 65.31× | 35.9% | 30.8% | 15.4% | 0.48× | 2.92× | 72.97× | 46.2% | 38.5% |
| Retrieval Prompt (HLStrans) | Deepseek-R1 | 25.6% | 0.21× | 2.10× | 19.01× | 48.7% | 46.2% | 33.3% | 0.41× | 2.19× | 20.40× | 53.8% | 56.4% |
| | GPT-5 | 25.6% | 0.19× | **37.10×** | 425.07× | 53.8% | 53.8% | **46.2%** | 0.46× | **37.10×** | 962.12× | **66.7%** | **71.8%** |
| | Grok-4 | 20.5% | 0.53× | 2.56× | 49.77× | **64.1%** | 51.3% | 30.8% | 0.57× | 2.84× | 51.85× | 51.3% | 56.4% |
| | Gemini-2.5-pro | **33.3%** | **0.98×** | 2.98× | 37.57× | 46.2% | 46.2% | 33.3% | 1.26× | 3.35× | 38.62× | 51.3% | 59.0% |
| | Qwen coder 32B | 15.4% | 0.37× | 1.910× | 5.87× | 61.5% | **59.0%** | 20.5% | 0.63× | 2.12× | 6.32× | 64.1% | 64.1% |

## 4.4 RESULTS ANALYSIS

**Observation 1. Retrieval-augmented generation and finetuning on HLStrans can improve model's performance on C to HLS task.** This demonstrates that our dataset by providing a rich cache of validated pragmas and code transformation examples serves as an indispensable "best practices" repository, steering LLMs toward hardware-friendly idioms and dramatically reducing synthesis failures.

**Observation 2. Sampling diversity substantially boosts results**. Allowing up to five candidate generations per input (Best@5) improves both synthesis success rate and achieved acceleration. For example, GPT-5 (retrieval prompt with HLStrans) raises synthesis accuracy from 53.8% to 71.8% under Best@5, underscoring the benefit of n-best generation.

**Observation 3. LLM optimization may harm HLS code performance.** We observe that some LLM-optimized kernels actually degrade performance (speedup $< 1\times$). This can happen for two reasons: First, the restructuring performed by the LLM can introduce new loop dependencies, increasing latency; Second, the pragmas inserted by LLMs may be less effective than the default optimizations inferred by the HLS compiler. Therefore, it is necessary to set up dataset to guide LLM's proper optimizations.

**Observation 4. Trade-off between pass rate and optimizations.** Applying retrieved, optimized code examples increases performance but reduces both functional and synthesis accuracy. For example, Deepseek-R1 (retrieval prompt with HLSdataset) increase speedup but decrease functional accuracy from 43.6% to 33.3% compared with zero-shot prompt. This highlights a trade-off between aggressive optimization and correctness when the LLM's capability is unchanged.

**Observation 5. LLMs perform differently across transformations.** As shown in Figure 1, pretrained models more easily apply T2 and T5. Fine-tuning on HLStrans improves the success rates of all the transformations, as reported in Table 2.

## 5 CONCLUSION

We introduce a novel dataset that transforms C or C++ kernels into richly annotated HLS implementations, empowering LLMs to learn hardware-aware optimizations such as loop pipelining, unrolling, and memory buffering. Our experiments demonstrate that retrieval and fine-tuning on this dataset significantly boosts both latency reduction and synthesis success rates, proving its effectiveness in accelerating and automating electronic design flows. By releasing the dataset and training scripts, we aim to catalyze further exploration at the intersection of LLMs and hardware design.

## ETHICS STATEMENT

We release a dataset that converts C/C++ kernels into richly annotated HLS implementations, together with training scripts, to accelerate LLM-driven hardware optimizations. While retrieval and fine-tuning improve latency and synthesis success, automated optimizations can produce incorrect or biased transformations; therefore the dataset and models are for research-only use and not intended for safety-critical deployment. Users should apply human review, evaluate functional correctness and synthesis safety alongside performance gains, and publish datasheets/model cards to promote transparency. Continued work on verification, robustness, and responsible reporting of failure cases is strongly encouraged.

## REPRODUCIBILITY STATEMENT

We are committed to ensuring the reproducibility of our findings. All datasets, code, and experimental scripts are publicly available at `https://anonymous.4open.science/r/HLStrans-B578/`.

## LLM USAGE DECLARATION

We used Gemini 2.5 Pro[3] to polish grammar and phrasing during the writing process. No part of the analysis, experimental design, or results was generated by a large language model.

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

# A APPENDIX

## A.1 HLS CODE TRANSFORMATIONS

### A.1.1 HLS CODE OPTIMIZATION: CODE RESTRUCTURING.

In our datasets, we apply a suite of code-reconstruction techniques designed to optimize memory access patterns, alleviate computational bottlenecks, and resolve loop dependencies. By restructuring data flow and exploiting hardware parallelism, these methods boost throughput and shorten overall execution time. Table 4 list the main Code Restructuring adopted in our dataset.

Table 4: HLS Code Reconstruction Methods

| Optimization | Explanation | Performance Benefit |
|---|---|---|
| Memory coalescing | Merge multiple memory accesses into one memory transaction. | Reduces memory access latency and improves bandwidth utilization. |
| Local tiling | Divide loops into tiles to improve cache reuse and spatial locality. | Enhances data locality and on-chip buffer efficiency. |
| Ping pong buffer | Alternate between two buffers for simultaneous load and compute. | Hides memory latency by overlapping computation with memory access. |
| Dataflow | Separate tasks into pipeline stages for concurrent execution. | Allows function-level parallelism, boosting throughput. |
| Control flow optimization | Replace if–else with ternary or simplified logic conditions. | Reduces combinational path length, improving timing and synthesis. |

### A.1.2 HLS CODE OPTIMIZATION: HLS DIRECTIVE (PRAGMA) INSERTION.

Our dataset features an extensive catalog of HLS pragmas ranging from memory-access directives (array partitioning, streaming) through loop-level transformations (unrolling, merging, tiling) to fine-grained pipeline controls (initiation interval tuning, dataflow regions). By systematically applying and combining these pragmas, these directives empower automated HLS flows to tailor synthesized hardware for domain-specific latency, throughput, and area requirements making our dataset a valuable reference for exploring pragma-driven performance tuning. Table 5 introduces these applied pragma optimizations.

Table 5: HLS Directive (Pragma) Insertion Methods

| Optimization | Explanation | Pragma Example |
|---|---|---|
| Array partition | Split a large array into multiple smaller memories | `#pragma HLS ARRAY_PARTITION variable=arr complete` |
| Memory type | Specify the on-chip storage type (BRAM/URAM/SMALL_RAM) | `#pragma HLS RESOURCE variable=buf core=RAM_2P` |
| Loop unroll | Replicate loop body to create parallel compute units | `#pragma HLS UNROLL factor=4` |
| Loop merge | Merge consecutive loops to reduce control overhead | `#pragma HLS LOOP_MERGE` |
| Function inline | Inline functions to eliminate call overhead | `#pragma HLS INLINE` |
| Pipeline | Pipeline loops or functions to lower initiation interval | `#pragma HLS PIPELINE II=1` |
| Dataflow | Enable task-level parallelism between functions | `#pragma HLS DATAFLOW` |
| Dependence | Declare data dependencies to allow safe loop optimizations | `#pragma HLS DEPENDENCE variable=arr inter false` |
| Stream | Use streaming interfaces to transfer data via FIFOs | `#pragma HLS STREAM variable=fifo depth=8` |

### A.1.3 HLS CODE OPTIMIZATION: DATA-TYPE ADAPTATION.

Our dataset also incorporates a comprehensive suite of data-type adaptations optimized for FPGA synthesis. We translate generic C types (e.g., `int`, `float`, `struct`) into precise HLS constructs such as `ap_uint<W>`, `ap_fixed<TOTAL,INT>`, and `hls::stream<T>` to fully exploit on-chip LUT/FF and DSP resources. These mappings enable fine-grained control over bit-width, data alignment, and streaming interfaces, ensuring maximal throughput, minimal logic utilization, and lower power consumption in FPGA deployments. Table 6 lists the specific datatype conversions applied in our framework.

Table 6: Adaptation of C Data Types to HLS Data Types

| Original C Type | HLS Type | Purpose |
|---|---|---|
| int/short/char | ap_uint<W>/ ap_int<W> | Precisely control integer bit-width to save LUT/FF resources |
| float/double | ap_fixed<TOTAL,INT>/ ap_ufixed<TOTAL,INT> | Replace floating-point with fixed-point to reduce DSP usage and power |
| struct/union | struct { ap_uint<...> field;} with bitfields | Precisely specify field bit-widths and alignment, eliminate padding |
| pointer/array | hls::stream<T> | Map to hardware FIFO streams for streaming transmission |

### A.1.4 HLS Code Optimization: Transformation of Functions

By transforming standard C/C++ functions into their corresponding HLS intrinsics, developers can leverage highly optimized FPGA kernels. This approach dramatically boosts execution performance by exploiting dedicated hardware units for math operations and data manipulation. At the same time, it conserves FPGA resources, reducing logic utilization and power consumption compared to generic software approximations. Table 7 lists the transformations of standard math functions to HLS intrinsics.

Table 7: Transformation of Standard Functions to HLS Intrinsics

| Standard C/C++ Function | HLS Intrinsic | Purpose |
|---|---|---|
| std::sqrt(x) | hls::sqrt(x) | Generates a pipelined square root unit instead of slow software approximation |
| std::exp(x) | hls::exp(x) | Synthesizes an exponential function hardware block (LUT-based) |
| std::log(x) | hls::log(x) | Provides a hardware-friendly implementation of natural logarithm |
| std::sin(x) | hls::sin(x) | Efficient sine computation using CORDIC or LUTs |
| std::cos(x) | hls::cos(x) | Efficient cosine computation using CORDIC or LUTs |
| a / b | hls::div(a, b) | Replaces division with a synthesizable divider core |
| a % b | hls::mod(a, b) | Synthesizes modulo operation in hardware |

### A.1.5 HLS Code Repair: HLS-Compliant Coding Style.

High-level synthesis (HLS) cannot synthesize all idiomatic C constructs directly. To enable hardware generation, we must refactor unsupported patterns like dynamic memory allocation, recursion, and pointer arithmetic into HLS-compliant coding styles that the tool can analyze and map to on-chip resources. Table 8 lists these common transformations.

Table 8: Transformation of Unsupported C Constructs for HLS Compatibility

| Unsupported C Construct | Recommended HLS-Compatible Transformation | Purpose |
|---|---|---|
| Dynamic memory allocations | Use static arrays with fixed size at compile time | HLS tools require compile-time memory size to synthesize physical resources (BRAM/LUTRAM) |
| Recursion | Convert to iterative form using for/while loops | Recursion creates a dynamic call stack, which is not synthesizable |
| Pointer arithmetic beyond array indexing | Use bounded array indexing | Allows compiler to infer memory access patterns and pipeline-optimize |
| Function pointers or callbacks | Inline or manually instantiate function variants | HLS requires all control flow to be static and analyzable at compile time |
| Variable-length arrays | Replace with fixed-size arrays defined by constants or macros | HLS cannot synthesize dynamically sized buffers |

## A.2 DATASET AUGMENTATION

### A.2.1 MCTS FRAMEWORK

MCTS enables an agent to learn to navigate the vast space of possible code transformations while balancing multiple optimization objectives. The agent's decisions are guided by comprehensive feedback from the HLS toolchain, including synthesis success, resource utilization, and performance metrics. Our MCTS has the following elements:

***Environment E:*** The environment is composed of the HLS toolchain, specifically Xilinx Vitis HLS, which compiles the code and provides critical feedback such as performance estimates.

***Agent G:*** We propose to use an LLM as the agent that leverages its pretrained knowledge of hardware design and in-context learning abilities.

***Action A:*** At each time step $t$, the agent selects an action $a_t$, which corresponds to a prompt or transformation applied to the current HLS code. We define two complementary action types: RAG-based actions retrieve optimization policies directly from our pre-built table and accompanying code examples shown in Figure 6, leveraging retrieval-augmented generation to surface proven transformations rapidly and reliably. Reasoning-based actions with ReAct prompt (Yao et al., 2023), in contrast, analyze compiler warnings such as pipeline-interval breaches or loop-unroll violations and apply targeted code reforms by interpreting warning semantics within the current code context.

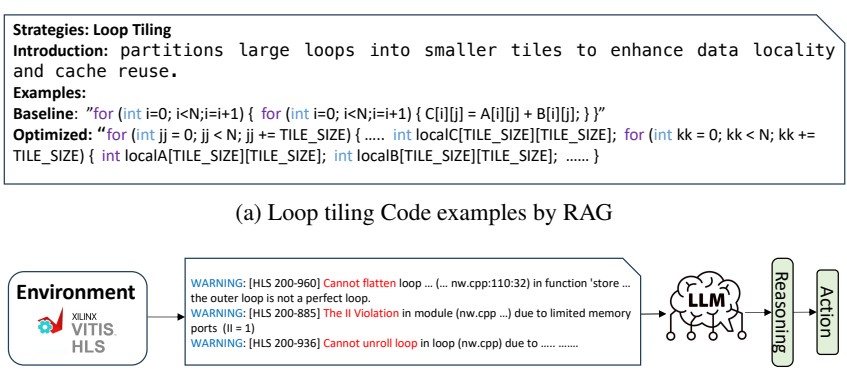

(a) Loop tiling Code examples by RAG

(b) LLM reasoning about environment warning and tool hint

Figure 6: Actions design of MCTS

***State S:*** The state $S_t$ at time step $t$ is defined as the current version of the HLS code after applying the previous actions.

***Reward R:*** Rule-based reward shaping has proven effective in guiding agent behavior in previous work Guo et al. (2025). In our framework, the reward function $R(s_t, a_t)$ is computed by applying rule-based scoring to verification results and feedback provided by the HLS tool. A penalty of $-2$ is applied when the verification fails, and $-1$ if synthesis fails or the design exceeds resource constraints. A neutral reward of $0$ is given when the transformed design performs worse than the original, while a reward of $1$ is granted when it performs better. If the design not only surpasses the original but also meets timing constraints, a higher reward of $2$ is assigned.

In our cases, the MCTS begins at the initial state $S_0$ (the root node), which is the naive HLS code. From a state $S_t$, the agent applies an optimization policy $\pi$, i.e., an action $a_t \in \mathcal{A}$, transitioning to the subsequent state $S_{t+1}$. MCTS consists of four key phases: selection, expansion, simulation, and backpropagation. $N(S_t)$, the total number of times $S_t$ has been visited.

**Selection**: We employ the upper confidence bounds for trees (UCT) (Gelly & Wang, 2006) algorithm to choose nodes.

$$\pi(s_t) = \arg\max_{a_t \in \mathcal{A}} \left( \underbrace{R(s_t, a_t)}_{\text{reward}} + \beta \times \underbrace{\frac{\sqrt{1 + N(s_t)}}{1 + N(s_t, a_t)}}_{U \text{ Term}} \right).$$

**Expansion:** From the current state $s_t$, generate one or more child nodes to explore untried actions.

**Simulation:** Perform a rollout from the chosen child node by applying $a_{t+1}$, running HLS synthesis to estimate latency and resource utilization, and computing the reward $R(s_t, a_{t+1})$.

**Backpropagation:** Propagate the obtained reward back up the visited path, updating each node's statistics (e.g., visit count and value estimate) to improve future selection.

**Retrieval-Augmented Generation:** To broaden the range of HLS code-transformation techniques that our LLM can learn, we built an automated framework (see Appendix A.2) that programmatically generates optimized variants via Monte Carlo Tree Search. Central to this system is a Retrieval-Augmented Generation (RAG) table of optimization strategies including code-reconstruction patterns, directive (pragma) insertions, data-type adaptations, and function-level transformations each entry pairing a concise description with a few-shot example illustrating the baseline code and its optimized counterpart listed in A.1. During search, these RAG-driven actions guide the MCTS policy to apply specific transformations, yielding a diverse corpus of HLS kernels ready for LLM fine-tuning and evaluation. One kind of Retrieval-Augmented strategies is shown in Figure 7 and prompt template is shown in Figure 8.

```
Strategies: Loop Tiling
Type: Need to refactor the code
Introduction: partitions large loops
into  smaller  tiles  to  enhance
data locality and cache reuse.
Examples:
Baseline: "for (int i=0; i<N;i=i+1) {
for (int i=0; i<N;i=i+1) { C[i][j] = A[i][j] + B[i][j]; } }"
Optimized: "for (int jj = 0; jj < N; jj += TILE_SIZE)
{ .....
     int localC[TILE_SIZE][TILE_SIZE];
     for (int kk = 0; kk < N; kk += TILE_SIZE) {
     int localA[TILE_SIZE][TILE_SIZE];
     int localB[TILE_SIZE][TILE_SIZE]; ...... }
```

Figure 7: Example of Retrieval-Augmented strategies in MCTS framework

```
You are a FPGA engineer, You should obey Xilinx HLS code guidelines. The name of top_function is
{function_name}, it can not be changed
        The code should have a header(h) file named {top_function}.h and a cpp file named
{top_function}.cpp The defination of variables, constants and functions are only in header file.
        In cpp file, you should firstly give sub functions of code, the codes of top function should
be at the end of cpp file.
        Your aim is to make sure the function of code is right and the pipeline interval from Xilinx
HLS log to be one to achieve better performance.
        You should optimize the following HLS code using these strategies:\n\n''' +
"\n".join(strategies)
```

Figure 8: Prompt Template for the optimization with MCTS framework

**Framework Evaluation:** We evaluated our dataset-augmentation framework on the widely adopted Rodinia benchmark suite (Che et al., 2009), using a Xilinx Alveo U55C FPGA board running at a 300 MHz kernel clock. Our goal was to measure how effectively our MCTS-based sampler could guide Deepseek-R1 and GPT-4o toward highly optimized HLS kernels.

Figure 9 shows the average success rate in the benchmarks. As the figure shows, their success rate is Qwen32B > Deepseek-R1 > GPT-4o > Qwen7B while Deepseek-R1 can achieve highest average speedup. The results show performance-increase will degrade the ability of LLM to produce the correct HLS code.

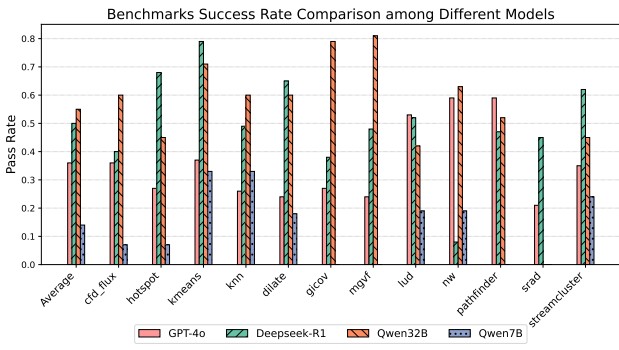

Figure 9: Success rate on different benchmarks with different models.

Table 9 summarizes the Best@1 kernel runtimes (in milliseconds) across twelve diverse applications, comparing four configurations: Baseline (The unmodified, compiler-generated HLS implementation), HLSPilot (Xiong et al., 2024) (A recent LLM based optimization framework), GPT-4o with our framework and Deepseek-R1 with our framework. Our results reveal several key findings:

- Consistent Improvement over previous work. In every benchmark, both of our enhanced pipelines outperform HLSPilot, demonstrating that the combination of large-model code generators with MCTS exploration yields more hardware-efficient HLS designs.
- Deepseek-R1 with our framework achieves up to average 28× reduction in real execution time compared to the baseline.GPT-4o with our framework attains up to average 20× reduction in real execution time.
- Robust Gains Across Diverse Kernels. From compute-bound codes such as kmeans, mgvf, and streamcluster, to memory-sensitive workloads like hotspot and nw, our framework consistently identifies and applies scheduling, pipelining, and data-partitioning transformations that exploit the parallelism and memory hierarchy of the Alveo U55C.

Table 9: Runtime (ms) of different benchmarks across models.

| Application | Baseline | HLSPilot Xiong et al. (2024) | Ours | |
|---|---|---|---|---|
| | | GPT-4o | GPT-4o | Deepseek-R1 |
| cfd_flux | 13 | 6.71 | 4.57 | 1.61 |
| hotspot | 1879.1 | 712.7 | 300.5 | 22.3 |
| kmeans | 2243.2 | 65.9 | 17.9 | 15.7 |
| knn | 17.0 | 2.8 | 0.83 | 0.82 |
| dilate | 48.8 | 16 | 0.75 | 1.64 |
| gicov | 107.0 | 93 | 82.3 | 30.7 |
| mgvf | 8047.5 | 3212 | 1231 | 446 |
| lud | 226.4 | 112 | 81.2 | 52.6 |
| nw | 206.4 | 145 | 73 | 13 |
| pathfinder | 7.8 | 5.9 | 1.09 | 1.51 |
| srad | 35.7 | 9.4 | 6.4 | 6.6 |
| streamcluster | 16173 | 9388 | 8162.3 | 3966 |

A.2.2 DESIGN SPACE EXPLORATION.

The tool is an automated HLS design-space explorer that employs a genetic-algorithm optimizer to discover effective directive combinations—specifically loop pipelining, loop unrolling, and array partitioning—that maximize performance and resource efficiency. We traverse the search space with the NSGA-II algorithm (Deb et al., 2002) as implemented in the `PyMOO` library (Blank & Deb, 2020), chosen for its ability to escape local optima and rapidly converge to high-quality solutions. NSGA-II is executed for 24 generations with a population size of 40. Each generation performs three steps: (1) generate or initialize the population, (2) apply each candidate configuration to the source code using compiler B2 and synthesize with `Xilinx Vitis`, and (3) return the synthesis metrics

to NSGA-II. Configurations that exceed device resources or demand prohibitive HLS runtimes (e.g., $\gtrsim$ 1 hour) are deemed infeasible and discarded. Genetic operators are configured as follows: random sampling/selection (mutation sampling probability = 0.1), simulated binary crossover (probability = 1.0, $\eta$ = 15), and polynomial mutation ($\eta$ = 20); all other operator parameters use `PyMOO` defaults.

### A.2.3 DATASET EXAMPLES

This section presents three real HLS code transformation pair examples: performance optimization (Code restructuring and Directive insertion), synthesizability correction (Code repair), and adaptation from C-style to HLS-style code (Data-type adaptation and transformation of functions). The dataset is hosted at `https://huggingface.co/datasets/qingyun777yes/HLStrans`.

**1. Performance Optimization** Figures 10a and 10b show a simple K-Nearest Neighbors (KNN) implementation before and after HLS optimization. The optimized version achieves better performance due to improved pipelining, parallelism, and memory optimization.

**2. Synthesizability Transformation** Figures 11a and 11b illustrate the transformation from a non-synthesizable function into a valid HLS-compatible version.

**3. C-style to HLS-style Conversion** Figures 12a and 12b demonstrate how C-style data types and functions can be adapted into HLS-friendly forms.

### A.3 PROMPT DETAILS

We explore three types of prompts used for HLS code transformation: zero-shot, chain-of-thought, and retrieval-augmented.

```
extern "C"{
void workload(
    float inputQuery[NUM_FEATURE],
    float searchSpace[NUM_PT_IN_SEARCHSPACE*NUM_FEATURE],
    float distance[NUM_PT_IN_SEARCHSPACE]
){
    #pragma HLS INTERFACE m_axi port=inputQuery offset=slave bundle=gmem
    #pragma HLS INTERFACE s_axilite port=inputQuery bundle=control
    #pragma HLS INTERFACE m_axi port=searchSpace offset=slave
bundle=gmem
    #pragma HLS INTERFACE s_axilite port=searchSpace bundle=control
    #pragma HLS INTERFACE m_axi port=distance offset=slave bundle=gmem
    #pragma HLS INTERFACE s_axilite port=distance bundle=control
    #pragma HLS INTERFACE s_axilite port=return bundle=control

    float sum;
    float feature_delta;
L1:  for(int i = 0; i < NUM_PT_IN_SEARCHSPACE; ++i){
        sum = 0.0;
L2:     for(int j = 0; j < NUM_FEATURE; ++j){
            feature_delta = searchSpace[i*NUM_FEATURE+j] - inputQuery[j];
            sum += feature_delta*feature_delta;
        }
        distance[i] = sum;
    }
    return;
}
}
```

(a) Unoptimized KNN implementation

```
...
void workload(
float inputQuery[NUM_FEATURE],
INTERFACE_WIDTH searchSpace[NUM_PT_IN_SEARCHSPACE*NUM_FEATURE/WIDTH_FACTOR],
INTERFACE_WIDTH distance[NUM_PT_IN_SEARCHSPACE/WIDTH_FACTOR]
){
#pragma HLS INTERFACE m_axi port=inputQuery offset=slave bundle=gmem
#pragma HLS INTERFACE s_axilite port=inputQuery bundle=control
#pragma HLS INTERFACE m_axi port=searchSpace offset=slave bundle=gmem
#pragma HLS INTERFACE s_axilite port=searchSpace bundle=control
#pragma HLS INTERFACE m_axi port=distance offset=slave bundle=gmem
#pragma HLS INTERFACE s_axilite port=distance bundle=control
#pragma HLS INTERFACE s_axilite port=return bundle=control

L7: float local_inputQuery[NUM_FEATURE];
L8: INTERFACE_WIDTH local_searchSpace_0[NUM_PT_IN_BUFFER*NUM_FEATURE/WIDTH_FACTOR];
L9: INTERFACE_WIDTH local_searchSpace_1[NUM_PT_IN_BUFFER*NUM_FEATURE/WIDTH_FACTOR];
L10: INTERFACE_WIDTH local_distance_0[NUM_PT_IN_BUFFER/WIDTH_FACTOR];
L11: INTERFACE_WIDTH local_distance_1[NUM_PT_IN_BUFFER/WIDTH_FACTOR];
L12: LOAD_INPUTQUERY: for (int i(0); i<NUM_FEATURE; ++i){
#pragma HLS UNROLL
local_inputQuery[i] = inputQuery[i];
}
L13: TILED_PE: for (int tile_idx(0); tile_idx<NUM_TILES+2; ++tile_idx){
#pragma HLS pipeline
int load_flag = tile_idx >= 0 && tile_idx < NUM_TILES;
int compute_flag = tile_idx >= 1 && tile_idx < NUM_TILES + 1;
int store_flag = tile_idx >= 2 && tile_idx < NUM_TILES + 2;
if (tile_idx % 2 == 0) {
load(load_flag, tile_idx, searchSpace, local_searchSpace_0);
compute(compute_flag, local_inputQuery, local_searchSpace_1, local_distance_1);
store(store_flag, tile_idx-2, local_distance_0, distance);
}
else {
load(load_flag, tile_idx, searchSpace, local_searchSpace_1);
compute(compute_flag, local_inputQuery, local_searchSpace_0, local_distance_0);
store(store_flag, tile_idx-2, local_distance_1, distance);
}
}
return;
}
}
```

(b) Optimized KNN implementation

Figure 10: Comparison of KNN implementations: (a) Unoptimized and (b) Optimized for high performance.

```c
#include <stdlib.h>
#include <stdio.h>
#include <string.h>
long long compute_sum(int *din, int N, int width) {
    long long *out_accum = malloc(sizeof(long long));
    int *array_local    = malloc(64 * sizeof(int));
    for (int i = 0; i < N; i++) {
        if (i < width) array_local[i] = din[i];
        else           array_local[i] = din[i] >> 2;
    }
    *out_accum = 0;
    for (int j = 0; j < N; j++) {
        *out_accum += array_local[j];
    }
    long long result = *out_accum;
    free(out_accum);
    free(array_local);
    return result;
}
```

(a) Original non-synthesizable code

```c
#include <stdlib.h>

long long compute_sum(int *din, int N, int width) {
#ifdef NO_SYNTH
    long long *out_accum = malloc(sizeof(long long));
    int *array_local    = malloc(64 * sizeof(int));
#else
    long long _out_accum;
    int _array_local[64];
    long long *out_accum = &_out_accum;
    int *array_local     = _array_local;
#endif

#pragma HLS ARRAY_PARTITION variable=_array_local complete
    LOOP_SHIFT: for (int i = 0; i < N; i++) {
#pragma HLS PIPELINE II=1
        array_local[i] = (i < width) ? din[i] : (din[i] >> 2);
    }
    *out_accum = 0;
    LOOP_ACCUM: for (int j = 0; j < N; j++) {
#pragma HLS PIPELINE II=1
        *out_accum += array_local[j];
    }
    return *out_accum;
}
```

(b) Modified synthesizable code

Figure 11: Transformation from non-synthesizable code to synthesizable code: (a) Original and (b) Modified version.

```
#include <math.h>
#include <ap_fixed.h>

#define C 64
#define H 28
#define W 28

void tanh(float input[C][H][W], float output[C][H][W])
{
  for (int c = 0; c < C; ++c) {
    for (int h = 0; h < H; ++h) {
      for (int w = 0; w < W; ++w) {
        output[c][h][w] = std::tanhf(input[c][h][w]);
      }
    }
  }
}
```

(a) Original C-style code using standard data types

```
...
typedef ap_fixed<16, 5> data_t;
void store_feature_map(data_t output_buffer[C][H][W], data_t
output_dram[C][H][W])
{
  #pragma HLS inline off
  for (int c = 0; c < C; c++)
  {
    for (int h = 0; h < H; h++)
    {
      for (int w = 0; w < W; w++)
      {
        output_dram[c][h][w] = output_buffer[c][h][w];
      }
    }
  }
}
void compute_exp(data_t input[C][H][W], data_t output[C][H][W])
{
  #pragma HLS inline off
  for (int i = 0; i < C; i++)
  {
    for (int j = 0; j < H; j++)
    {
      for (int k = 0; k < W; k++)
      {
        output[i][j][k] = hls::exp(input[i][j][k]);
      }
    }
  }
} ...
```

(b) Transformed HLS-style code using synthesizable types

Figure 12: Transformation from traditional C-style to HLS-style coding: (a) Original code and (b) Synthesizable HLS code.

If the program can not be synthesized, please turn it into synthesizable codes. If it is a slow high level synthesis FPGA program, optimize their performance with minimal resource.

 ### program : {src_code}

Must Only return the code use the format. \n
Example response format:

    ```cpp \n
    // implementation content here
    ```

Figure 13: Zero-shot prompt used for HLS code transformation.

```
Let's think step by step to optimize the HLS code.
Example 2:
Q: This is a slow HLS FPGA program. Please optimize it with array partitioning and loop unrolling to improve parallelism.
```cpp
void vector_add(const int A[32], const int B[32], int C[32]) { for (int i = 0; i < 32; i++) {  C[i] = A[i] + B[i]; } }
```
1. Identify memory contention: single-port arrays limit one access per cycle.
2. Partition arrays: use #pragma HLS ARRAY_PARTITION variable=A/B/C cyclic factor=4 to split each into 4 banks for parallel access.
3. Unroll the loop: add #pragma HLS UNROLL factor=4 so 4 additions execute in one cycle, matching the 4-way partition.
4. Keep pipelining: you may optionally add #pragma HLS PIPELINE II=1 for consistency.
```cpp
void vector_add(const int A[32], const int B[32], int C[32]) {
#pragma HLS ARRAY_PARTITION variable=A cyclic factor=4\n #pragma HLS ARRAY_PARTITION variable=B cyclic factor=4
#pragma HLS ARRAY_PARTITION variable=C cyclic factor=4\n for (int i = 0; i < 32; i++) { #pragma HLS UNROLL factor=4
        C[i] = A[i] + B[i]; } }
```
Now apply the same step-by-step reasoning to the following slow HLS code and provide the fully annotated, optimized version:'''
If the program can not be synthesized, please turn it into synthesizable codes. If it is a slow high level synthesis FPGA program, optimize their performance with minimal resource.
 ### program : {src_code}
Must Only return the code use the format. \n
Example response format:

    ```cpp \n
    // implementation content here
    ```
```

Figure 14: Chain-of-thought prompt for step-by-step transformation.

## A.4  DETAILED EXPERIMENT RESULTS

Unlike traditional software code, which need only pass functional correctness tests, HLS-generated kernels must also successfully synthesize and implement via the Vitis HLS toolchain to be deploy-

Let's think step by step to optimize the HLS code.
Q: This is a slow HLS FPGA program, which is similar to the current unoptimized codes.
Retrieval codes {Retrieval codes }

Now apply the same step-by-step reasoning to the following slow HLS code and provide the fully annotated,
optimized version:'''
If the program can not be synthesized, please turn it into synthesizable codes. If it is a slow high level synthesis
FPGA program, optimize their performance with minimal resource.
 ### program : {src_code}
Must Only return the code use the format. \n
Example response format:
   ```cpp \n
   // implementation content here
   ```

Figure 15: Retrieval-augmented prompt for enhanced transformation.

able on FPGA hardware. Below, we briefly describe how we leverage synthesis results for design
evaluation. Also, we introduce the fine tuning results during training.

### A.4.1  HLS SYNTHESIS RESULTS EXAMPLE

While HLS synthesis cannot yield perfectly accurate timing or resource-utilization numbers, it pro-
vides essential estimates for comparing design variants. Figures 16a and 16b show the synthesis re-
ports for the unoptimized and optimized KNN kernels, respectively, targeting a Xilinx Alveo U55C
accelerator at a 300 MHz kernel clock.

The optimized design trades increased resource usage more DSP slices, flip-flops (FFs), and lookup
tables (LUTs) for a dramatic fourfold reduction in latency cycles (from 2,097,324 cycles down to
508,479 cycles). In a real FPGA deployment, this corresponds to an end-to-end runtime of ap-
proximately 3.2 ms versus roughly 17 ms for the unoptimized kernel, while still remaining within
resource budgets. We report *acceleration* based on the estimated latency cycles from the synthe-
sis reports. However, HLS synthesis itself can be time-consuming, particularly for large or highly
optimized designs.

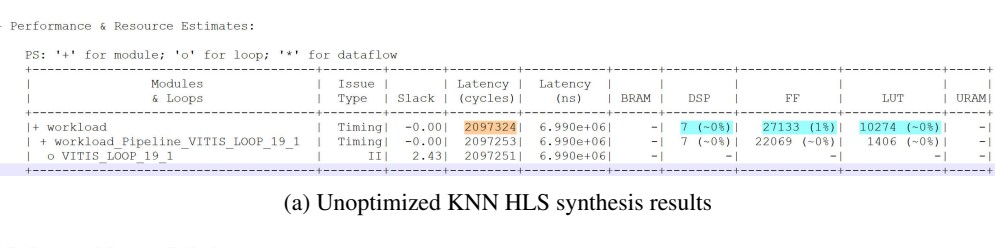

(a) Unoptimized KNN HLS synthesis results

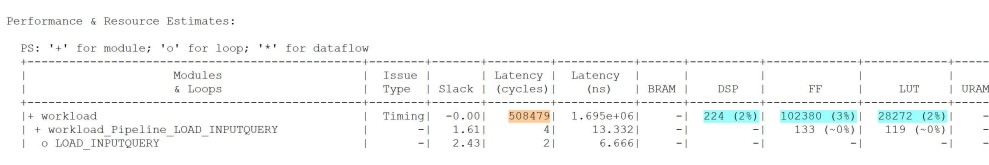

(b) Optimized KNN HLS synthesis results

Figure 16: (a) Unoptimized and (b) optimized KNN implementations after HLS synthesis.

### A.4.2 FINETUNE RESULTS

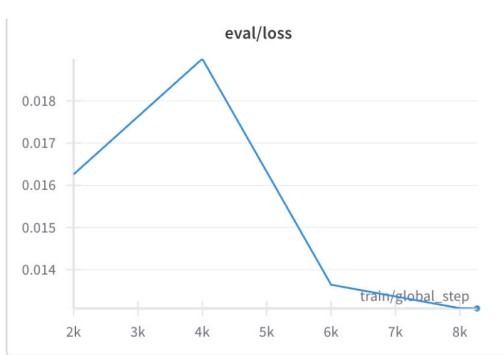 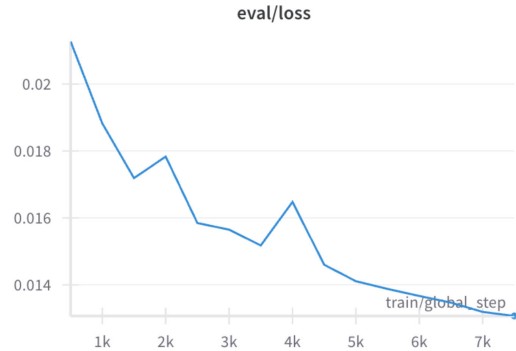

(a) Validation results for Qwen2.5-Coder-3B-Instruct during fine-tuning.

(b) Validation results for Qwen2.5-Coder-7B-Instruct during fine-tuning.

Figure 17: Validation performance of Qwen2.5-Coder models during fine-tuning on the held-out dataset.

With our real-world corpus, we reserve two C programs for the repair task and another 39 programs for the optimization task. The remaining 270 programs are split into training and validation sets. Figure 17a presents the validation loss curve for Qwen2.5-Coder-3B-Instruct, while Figure 17b shows the corresponding curve for Qwen2.5-Coder-7B-Instruct during fine-tuning. In both cases, the steadily decreasing loss demonstrates that fine-tuning effectively adapts the models to our dataset.

### A.5 IMPACT OF C TO HLS TASK

While HLS is syntactically close to C, we believe the task has the following meaning.

**Impact for reducing performance gap.** HLS is designed to accelerate hardware design, but there remains a substantial gap between plain C code and high-performance HLS code. In our experiments, LLM-generated samples can achieve up to hundreds of speedup over the original code.

**Impact for reducing coding budget.** To obtain high-quality HLS code, developers must perform non-trivial semantic transformations—such as loop tiling, bitwidth narrowing, converting buffer-based designs to streaming, or repairing code to satisfy synthesis constraints. These transformations are time-consuming and require HLS expertise. A fine-tuned LLM can automate or assist with many of these steps, significantly reducing development effort and turnaround time.

**Impact for agile hardware design with HLS.** Agile hardware design that starts from HLS enables software engineers to develop hardware accelerators more easily. However, understanding the hardware-specific transformations required for optimization is non-trivial. Our dataset and fine-tuned LLM help software engineers better design hardware accelerators.

**Real Case study: C to HLS task.** We use a genomics application as a real-world case study for the C-to-HLS conversion task Cong et al. (2022) in Table 10. High-performance HLS implementations include several components Cong et al. (2022). Converting C to HLS consumes 41% of the engineering effort, covering compiler directives, double buffering, and related transformations, whereas the function-level C code accounts for 59%. These conversion steps can require days to finish Cong et al. (2022), indicating that C-to-HLS conversion is a challenging problem that merits deeper study.

### A.6 TRANSFERRING ON DIFFERENT PLATFORMS

To clarify our claim: constructing high-performance HLS implementations from C typically requires the five transformations illustrated in Figure 1. These transformations are common across modern HLS toolchains such as Vitis HLS, SmartHLS, and Bambu HLS. Table 11 lists some examples of five transformations for different HLS tools including SmartHLS, Vitis HLS and Bambu HLS.

Table 10: Breakdown of effort for a real-world C-to-HLS conversion task.

| Category | Sub-category | LOC | Percentage |
|---|---|---|---|
| **Functionality code** | | **308** | **59%** |
| **Optimizations code** | | **216** | **41%** |
| **Optimizations code** | Compiler directives | 48 | 22% |
| | Double buffering | 46 | 21% |
| | Frequency optimization | 38 | 18% |
| | PE duplication | 32 | 15% |
| | Others | 52 | 24% |

Table 11: Common HLS transformations and examples in different toolchains

| Transformation | Why needed (in HLS) | Vitis HLS example | SmartHLS example | Bambu HLS example |
|---|---|---|---|---|
| T1: Code Restructuring | Expose data locality and so on | loop tiling, dataflow | loop tiling, dataflow | loop tiling, dataflow |
| T2: Directive (Pragma) Insertion | Increase parallelism and so on | #pragma HLS UNROLL | #pragma HLS loop unroll | #pragma HLS unroll |
| T3: Data-Type Adaptation | Adapt to platform | `ap_int<64>` | `ap_int<64>` | `ap_int<64>` |
| T4: Transformation of Functions | Hardware implementations for expensive math or others | `sqrt` | `sqrt` | `sqrt` |
| T5: HLS-Compliant Coding Style | Recursion or dynamic memory allocation not synthesizable | recursion | recursion | recursion |

The augmentation techniques and the benchmarking methodology operate at the level of HLS transformations and therefore generalize across modern HLS toolchains. To substantiate this claim, we evaluate the generality of our approach on two additional HLS toolchains: Bambu and SmartHLS (LegUp).

We apply our augmentation pipeline to transform C programs into high-performance HLS designs by performing the five targeted transformations described in Figure 1. For each benchmark/toolchain we report two metrics: *Speedup*, the relative performance improvement of the optimized design over the baseline; and *Pass rate*, the fraction of generated designs that both pass the functional tests and successfully synthesize. These results in Table 12 and 13 demonstrate that our augmentation techniques produce measurable performance gains across multiple, independently developed HLS toolchains, supporting the claim that the pipeline and evaluation methodology generalize beyond a single vendor.

Table 12: Augmentation pipeline evaluation for Bambu HLS

| Metric | cfd_flux | dilate | gicov | hotspot | kmeans | knn | nw | pathfinder | srad | streamcluster |
|---|---|---|---|---|---|---|---|---|---|---|
| Speedup | 2.31 | 24.5 | 1.06 | 4.10 | 41.7 | 9.52 | 2.72 | 3.36 | 2.91 | 1.20 |
| Pass rate | 0.29 | 0.19 | 0.21 | 0.21 | 0.30 | 0.20 | 0.47 | 0.46 | 0.16 | 0.27 |

Table 13: Augmentation pipeline evaluation for SmartHLS

| Metric | cfd_flux | dilate | gicov | hotspot | kmeans | knn | nw | pathfinder | srad | streamcluster |
|---|---|---|---|---|---|---|---|---|---|---|
| Speedup | 2.8 | 30.1 | 1.3 | 5.2 | 50.1 | 11.6 | 3.4 | 4.2 | 3.7 | 1.5 |
| Pass rate | 0.36 | 0.24 | 0.27 | 0.27 | 0.37 | 0.26 | 0.59 | 0.59 | 0.21 | 0.35 |

We evaluate our benchmarks on two additional HLS toolchains, Bambu HLS and SmartHLS (LegUp), using multiple LLMs. Table 14 and Table 15 report the zero-shot best@1 prompting results and error breakdown for Bambu HLS; Table 16 and Table 17 provide the corresponding results for SmartHLS. Metrics are defined in Section 4.2 of the manuscript. "Speed/Opt" denotes the fraction of cases with any improvement (reported as percentage), "Min/Avg/Max" are relative speedups, "Functional Accuracy" is the fraction of outputs passing functional tests, and "Synthesis Accuracy" is the fraction that both pass functional tests and successfully synthesize.

Table 14: Benchmark results of Bambu HLS

| Model | Opt (%) | Min | Avg | Max | Functional Accuracy | Synthesis Accuracy |
|---|---|---|---|---|---|---|
| Deepseek-R1 | 12.8% | 0.10× | 1.10× | 10.2× | 30.8% | 28.2% |
| GPT-5 | 15.4% | 0.03× | 8.20× | 310.5× | 33.3% | 33.3% |
| Grok-4 | 12.8% | 0.30× | 1.50× | 30.3× | 28.2% | 28.2% |
| Gemini-2.5-pro | 17.9% | 0.60× | 1.90× | 21.2× | 25.6% | 25.6% |
| Qwen coder 32B | 10.3% | 0.20× | 0.70× | 2.5× | 38.5% | 35.9% |

Table 15: Error analysis of Bambu HLS.

| Model | Compiler Errors (%) | Output Errors (%) | Runtime Exceptions (%) | Resource Errors (%) | Directive Errors (%) |
|---|---|---|---|---|---|
| 32B | 40 | 8 | 15 | 15 | 22 |
| Deepseek-R1 | 41 | 9 | 17 | 13 | 20 |
| Gemini25 | 52 | 8 | 8 | 15 | 17 |
| GPT-5 | 34 | 11 | 21 | 11 | 23 |

Table 16: Benchmark results of SmartHLS (LegUp).

| Model | Opt (%) | Min | Avg | Max | Functional Accuracy | Synthesis Accuracy |
|---|---|---|---|---|---|---|
| Deepseek-R1 | 15.4% | 0.08× | 0.90× | 9.0× | 25.6% | 23.1% |
| GPT-5 | 12.8% | 0.02× | 7.50× | 200.0× | 35.9% | 35.9% |
| Grok-4 | 10.3% | 0.25× | 1.20× | 25.0× | 23.1% | 25.6% |
| Gemini-2.5-pro | 15.4% | 0.70× | 2.10× | 25.0× | 28.2% | 25.6% |
| Qwen coder 32B | 12.8% | 0.25× | 0.80× | 3.0× | 33.3% | 38.5% |

Table 17: Error analysis of SmartHLS.

| Model | Compiler Errors (%) | Output Errors (%) | Runtime Exceptions (%) | Resource Errors (%) | Directive Errors (%) |
|---|---|---|---|---|---|
| 32B | 38 | 9 | 16 | 14 | 23 |
| Deepseek-R1 | 39 | 10 | 16 | 12 | 23 |
| Gemini25 | 43 | 7 | 9 | 14 | 27 |
| GPT-5 | 33 | 12 | 20 | 12 | 23 |

## A.7 TESTBENCH GENERATIONS

We report the coverage results, lines, branches, tokens, and calls collected from `gcov` for our dataset, as shown in Table 18. While full (100%) coverage is not attainable, the table demonstrates that our testbench nevertheless yields robust, high-quality coverage for evaluation.

Table 18: Coverage results collected from `gcov` for our dataset.

| Range | Lines (%) | Branches (%) | Tokens (%) | Calls (%) |
|---|---|---|---|---|
| 100% | 94.82 | 94.82 | 79.29 | 92.88 |
| [75%,100%) | 4.85 | 4.85 | 10.03 | 0.65 |
| [50%,75%) | 0.00 | 0.32 | 9.71 | 6.47 |
| [25%,50%) | 0.32 | 0.00 | 0.97 | 0.00 |
| < 25% | 0.00 | 0.00 | 0.00 | 0.00 |

## A.8 CODE STRUCTURE ANALYSIS

For the code structure analysis we computed per-sample statistics including lines of code (LoC), number of functions, number of loops, and cyclomatic complexity in Table 19.

From these tables, we conclude that the dataset covers a wide variety of code styles and complexity levels, and is therefore appropriate for evaluating LLM performance on HLS-related tasks.

Table 19: Dataset distributions for code-structure metrics. Each cell shows the bin range (top) and the percentage of samples falling in that bin (bottom).

| Metric | Bin1 (Range) | Bin2 (Range) | Bin3 (Range) | Bin4 (Range) | Bin5 (Range) |
|---|---|---|---|---|---|
| Lines of Code (LoC) | [3.00, 44.40] 39.51% | [44.40, 85.80] 32.33% | [85.80, 127.20] 10.35% | [127.20, 168.60] 10.22% | [168.60, 210.00] 7.60% |
| Function number | [0.00, 2.00] 57.74% | [2.00, 4.00] 27.84% | [4.00, 6.00] 8.99% | [6.00, 8.00] 3.07% | [8.00, 10.00] 2.36% |
| Loop number | [0.00, 7.80] 41.88% | [7.80, 15.60] 27.34% | [15.60, 23.40] 15.79% | [23.40, 31.20] 10.75% | [31.20, 39.00] 4.23% |
| Cyclomatic complexity | [1.00, 14.00] 48.95% | [14.00, 27.00] 27.58% | [27.00, 40.00] 11.60% | [40.00, 53.00] 6.23% | [53.00, 66.00] 4.63% |

## A.9 EXPERIMENTAL ANALYSIS

### A.9.1 DETAILED ERROR ANALYSIS

We perform a fine-grained analysis of the failures produced by LLM-generated HLS designs and identify five dominant error categories: *Compiler Errors*, *Directive Errors*, *Runtime Exceptions*, *Resource Errors*, and *Output Errors*. Across both Bambu HLS and SmartHLS (LegUp), directive-related errors are particularly prevalent: models commonly emit Vitis-style pragmas even when the target tool requires a different pragma syntax. We attribute this behavior to the relative abundance and higher quality of Vitis HLS examples in training data.

**Compiler Errors**. These errors reflect syntactic or structural problems that prevent the HLS front-end from accepting the program (e.g., malformed C, undefined identifiers, or unsupported language constructs). Because such errors occur before downstream HLS passes, they represent a primary bottleneck in the overall workflow and indicate fragile tool compatibility.

**Directive Errors.** This category captures incorrect or unsupported pragma usage (e.g., wrong pragma names, invalid parameters, incorrect placement, or mixing pragmas intended for different tools). Directive errors show that models lack fine-grained tool-awareness: even small syntax

differences between toolchains (Vitis vs. Bambu vs. LegUp/SmartHLS) cause a large fraction of failures.

**Runtime Exceptions.** A nontrivial fraction of generated programs compile but fail during simulation (exceptions, timeouts, memory faults, or sandbox interruptions). These failures indicate difficulties in producing correct hardware control-path logic and robust testable code, beyond purely numerical computation.

**Resource Errors.** Resource-related failures occur when aggressive transformations (e.g., excessive unrolling or partitioning) push designs beyond the target device's resource budgets. Although less frequent than compiler or directive errors, resource errors are critical for practical deployability and show that models tend to over-parallelize without awareness of device constraints.

**Output Errors.** Semantic mismatches (wrong algorithmic behavior, off-by-one/boundary mistakes, or incorrect output format) are the least common error type. This suggests that, once a design compiles and simulates, LLMs generally preserve core algorithmic behavior reasonably well — i.e., functional correctness is easier to achieve than tool-specific syntactic and compilation constraints.

### A.9.2 SPEEDUP ANALYSIS

We analyze how model-generated transformations affect performance, focusing on the two stages with the largest impact: **T2** (pragma/directive insertion) and **T1** (code restructuring). Below we report the empirical distribution of optimization actions extracted from generated programs and summarize observed performance patterns.

**Breakdown of T2 (pragma / directive insertion):** Table 20–22 summarize the relative proportion of common T2 actions observed for each toolchain. Note that proportions reflect the fraction of generated designs that include a given action; a single design may include multiple actions, so row sums can exceed 100%.

Table 20: T2 (Vitis_HLS) distribution of pragma/directive actions (proportions).

| Action | Pragmas | Array-part | MemType | Unroll | Merge | Inline | Pipeline | Dataflow | Dep/Stream |
|---|---|---|---|---|---|---|---|---|---|
| Proportion | 43.6% | 12.8% | 53.8% | 5.1% | 28.2% | 82.1% | 10.3% | 10.3% | 10.3% |

Table 21: T2 (Bambu HLS) distribution of pragma/directive actions (proportions).

| Action | Pragmas | Unroll | Inline | Dataflow / Cache |
|---|---|---|---|---|
| Proportion | 30.8% | 69.2% | 10.3% | 87.2% |

Table 22: T2 (SmartHLS) distribution of pragma/directive (proportions).

| Action | Pragmas | Unroll | Inline | Dataflow | Pipeline / Partition |
|---|---|---|---|---|---|
| Proportion | 53.8% | 17.9% | 10.3% | 84.6% | 43.6% |

**Breakdown of T1 (code restructuring):** Table 23 reports the observed distribution of common T1 restructuring patterns. These transformations are closely related to memory-bound performance improvements.

Table 23: T1 code restructuring distribution.

| Action | Memory coalescing | Local tiling | Ping-pong buffer | Dataflow | Control-flow opt. |
|---|---|---|---|---|---|
| Percentage | 0.0% | 23.1% | 7.7% | 2.6% | 28.2% |

### A.9.3 OBSERVATIONS

- **Optimization can degrade performance.** Some LLM-generated transformations yield $<1\times$ speedup. Two common causes are (i) restructuring that introduces additional loop dependencies (increasing latency), and (ii) pragmas that are less effective than the tool's default optimizations. This observation underscores the need for dataset and reward signals that encourage *correct* (tool-aware) optimizations rather than aggressive but counterproductive rewriting.
- **T1 correlates with memory-bound gains.** For memory-intensive kernels, speedups are primarily driven by T1 transformations that improve memory behavior: memory coalescing (better burst efficiency), local tiling (reduced off-chip bandwidth), and ping-pong buffering (overlap of compute and memory).
- **T2 impact depends on application class.** Pipeline and Dataflow pragmas are most beneficial for streaming and stencil kernels where concurrency is the bottleneck. Unroll and Partition pragmas are critical for compute-bound kernels (e.g., KNN, GEMM). Inline and loop-merge transformations matter more in control-heavy applications by reducing scheduling overhead and enabling deeper pipelining.
- **Tool-specific defaults shape effectiveness.** The observed T2 distribution differs across toolchains because each HLS tool applies different default transformations and heuristics; consequently, identical pragma insertions can produce different outcomes across tools. This further motivates our claim that benchmarking at the transformation level (rather than at a single tool's syntax) yields more robust conclusions.

