# OpenReview forum: "HLStrans: Dataset for C-to-HLS Hardware Code Synthesis"
_ICLR.cc/2026/Conference — Submitted to ICLR 2026_

### Official Review · Reviewer_HLdE · 2025-10-31

**Soundness:** 3
**Presentation:** 3
**Contribution:** 3
**Rating:** 8
**Confidence:** 4

**Summary:**

The paper introduces HLStrans, a dataset containing 124K paired C/HLS programs for training LLMs to transform C code into optimized High-Level Synthesis (HLS) code for FPGAs. The dataset is created by collecting 309 base programs from various sources and augmenting them using an automated pipeline combining LLMs, Monte Carlo Tree Search (MCTS), and Design Space Exploration (DSE). The authors benchmark various LLMs showing improvements from retrieval and fine-tuning.

**Strengths:**

(1) Target is real and timely: LLMs for C→HLS need paired before/after code plus testbenches.
(2) Dataset is well-scaffolded around real HLS transformations, which is closer to what actual HLS engineers do than “just insert a pragma.”
(3) They try to close the loop with EDA feedback (Vitis), which is the right direction for LLM-in-the-loop hardware optimization.

**Weaknesses:**

(1) The augmentation relies entirely on DeepSeek-R1 and automated synthesis. There is no mention of expert validation of the generated samples' quality or correctness.
(2) All synthesis on Xilinx Alveo U55C, and thus it is unclear if insights transfer to other FPGAs (Intel, Lattice, etc.).

**Questions:**

(1) Please make an explicit comparison to Forgebench (2025): what does HLStrans offer that Forgebench doesn’t, besides testbenches? Do you cover more transformation types, or just organize them differently?

---

> ### Author Response · Authors · 2025-11-21
> **Authors' response for weakness 1**
>
> Thank you for taking the time to review our paper and for your detailed and constructive feedback. Please see our responses below;
>
> ### ***Weakness 1: Validation of the generated samples' quality and correctness.***
> We appreciate the reviewer’s observation. Augmented samples are evaluated using a manually written testbench and the synthesis tool in our flow. The generated samples must pass synthesis and the testbench, and must also surpass the original kernel’s performance.  The evaluation results are provided in the **Appendix A.2**; we  apologize for not making them more prominent.
>
> For correctness evaluation, I added the verification results as **Figure 7 in Appendix A.2** of the revised manuscript. The following table is the original data for Figure 7.
>
> | Model        | Average | cfd_flux | hotspot | kmeans | knn | dilate | gicov | mgyf | lud | nw | pathfinder | srad | streamcluster |
> |--------------|--------:|---------:|--------:|-------:|----:|--------:|------:|------:|-----:|----:|------------:|------:|----------------:|
> | **GPT-4o**       | 0.36 | 0.36 | 0.27 | 0.37 | 0.26 | 0.24 | 0.28 | 0.24 | 0.53 | 0.59 | 0.59 | 0.21 | 0.35 |
> | **Deepseek-R1**  | 0.50 | 0.40 | 0.68 | 0.79 | 0.49 | 0.65 | 0.38 | 0.48 | 0.52 | 0.08 | 0.47 | 0.45 | 0.62 |
> | **Qwen32B**      | 0.55 | 0.60 | 0.45 | 0.71 | 0.60 | 0.60 | 0.79 | 0.81 | 0.42 | 0.63 | 0.52 | 0.00 | 0.45 |
> | **Qwen7B**       | 0.14 | 0.07 | 0.07 | 0.33 | 0.33 | 0.18 | 0.27 | 0.00 | 0.19 | 0.19 | 0.00 | 0.00 | 0.24 |
>
>
> For quality evaluation, we showed the detailed results of Appendix A.2 on the widely used Rodinia benchmark suite [1], using a Xilinx Alveo U55C FPGA board running at a **300 MHz** kernel clock. Our goal was to measure how effectively our MCTS-based sampler could guide LLMs toward highly optimized HLS kernels. The table below summarizes the **Best@1 kernel runtimes (in milliseconds)** across twelve diverse applications, comparing four configurations:
>
> - Baseline (the unmodified HLS implementation)
>
> - HLSPilot (GPT-4o) (a recent LLM-based HLS code generation framework using GPT-4o)
>
> - Ours (GPT-4o) (our pipeline using GPT-4o)
>
> - Ours (Deepseek-R1) (our pipeline using Deepseek-R1)
>
> | Application   | Baseline | HLSPilot (GPT-4o) | Ours (GPT-4o) | Ours (Deepseek-R1) |
> |---------------|---------:|------------------:|--------------:|-------------------:|
> | cfd_flux      | 13.0     | 6.71              | 4.57          | **1.61**           |
> | hotspot       | 1879.1   | 712.7             | 300.5         | **22.3**           |
> | kmeans        | 2243.2   | 65.9              | 17.9          | **15.7**           |
> | knn           | 17.0     | 2.8               | 0.83          | **0.82**           |
> | dilate        | 48.8     | 16.0              | **0.75**      | 1.64               |
> | gicov         | 107.0    | 93.0              | 82.3          | **30.7**           |
> | mgvf          | 8047.5   | 3212.0            | 1231.0        | **446**            |
> | lud           | 226.4    | 112.0             | 81.2          | **52.6**           |
> | nw            | 206.4    | 145.0             | 73.0          | **13**             |
> | pathfinder    | 7.8      | 5.9               | 1.09          | 1.51               |
> | srad          | 35.7     | 9.4               | **6.4**       | 6.6                |
> | streamcluster | 16173.0  | 9388.0            | 8162.3        | **3966**           |
>
> According to the table's results,  our augmentation has **substantial speedups:** Our framework with Deepseek-R1 achieves an average **~28×** reduction in real execution time compared to the baseline, while GPT-4o achieves an average **~20×** reduction, but HLSpilot only achieves an average **~5×** reduction.
>
> [1] Chenwei, et al. "HLSpilot: LLM-based High-Level Synthesis.” ICCAD 2024.

---

> > ### Author Response · Authors · 2025-11-21
> > **Authors' response for weakness 2 and question**
> >
> > ### ***Weakness 2: Insights transfer to other platform***
> >
> > We agree that evaluating on a single device (Xilinx Alveo U55C) may limit the generality of our findings across different FPGA vendors. However, our methodology and insights are largely hardware-agnostic, because:
> >
> > - **High level optimization**: The optimization techniques we study (e.g., HLS-level transformations, kernel restructuring, memory-access optimizations) **operate above the vendor-specific backend**, which are common for other FPGA platforms.
> >
> > - **Observation**：Some findings are explored for language-model for code optimization research. For example, increasing sampling diversity often improves results, while blindly applying LLM-driven optimization can sometimes degrade HLS code quality. These observations can inspire other LLM-based code optimization framework.
> >
> > - ***Extensible to other HLS programs:*** We demonstrate portability by adapting our data augmentation pipeline to optimize FPGA workloads targeting **Microchip** devices (MPF100T-FCVG484I). The results are shown below:
> >
> > | Metric  | cfd_flux | dilate | gicov | hotspot | kmeans | knn  | nw  | pathfinder | srad | streamcluster |
> > | ------- | -------- | ------ | ----- | ------- | ------ | ---- | --- | ---------- | ---- | ------------- |
> > | Speedup | 2.8      | 30.1   | 1.3   | 5.2     | 50.1   | 11.6 | 3.4 | 4.2        | 3.7  | 1.5           |
> >
> > We believe that the proposed LLM-in-the-loop framework provides a reusable foundation for dataset generation and optimisation across HLS flows and FPGA platforms, rather than being restricted to a single tool. Researchers can directly reuse the components or adapt them to their own toolchains with minimal effort.
> >
> > ### ***Questions:  Compared with Forgebench***
> >
> >  Forgebench is future-HLS-ready, including a benchmark suite highlighting the necessity of HLS for a machine learning application.  As 3.4 HLStrans statistics show, our dataset has the following differences in the following tables. Also, the following table illustrates the number of samples for each transformation (T1, T2, T3, T4, T5) in our datasets.
> >
> > | Item                                                         | Our work (HLStrans)           | Forgebench | Notes                                              |
> > |--------------------------------------------------------------:|------------------------------:|------------------:|----------------------------------------------------|
> > | Transformation coverage                    | **T1, T2, T3, T4, T5**             | T1(Partly), T2, T3, T4 | HLStrans support more C to HLS transformation |
> > | Dataset Format                                   | **C and HLS**                     | HLS samples | HLStrans start with C to get an optimized HLS kernels
> > | Dataset size (number of entries)         | **124000**                             | 6000        |HLStrans are not limited to machine learning                                           |
> > | Application types                                 | **Different Kinds of applications**             | Machine learning               |  HLStrans are not limited to machine learning  |
> >
> > T1 restructures code to expose pipelining and dataflow through loop tiling, memory optimization, and parallel control flow. T2 adds pragmas (pipeline, dataflow, interface, etc.) to fine-tune scheduling and performance. T3 replaces generic types with bit-accurate HLS types (fixed point, ap_int) to save resources while meeting precision. T4 converts standard functions into HLS kernels or intrinsics to better exploit hardware accelerators. T5 enforces synthesis-friendly coding by removing unsupported constructs and using static arrays, simple loops, and explicit communication. The detailed information about transformation is in Appendix A.1 and Figure 1.
> >
> > [1] Shuai Che, Michael Boyer, Jiayuan Meng, David Tarjan, Jeremy W. Sheaffer, Sang-Ha Lee, and Kevin Skadron. Rodinia: A benchmark suite for heterogeneous computing. In 2009 IEEE International Symposium on Workload Characterization (IISWC), pp. 44–54, 2009. doi:10.1109/IISWC.2009.5306797.
> >
> > [2] Wanna A, Chen H, Hao C. ForgeBench: A Machine Learning Benchmark Suite and Auto-Generation Framework for Next-Generation HLS Tools[J]. arXiv preprint arXiv:2504.15185, 2025.

---

> > > ### Author Response · Authors · 2025-11-26
> > >
> > > Dear reviewer:
> > >
> > > We hope that you are satisfied with our answers and the additional results we have provided. As the discussion period comes to an end, we would be grateful if you could let us know if we have adequately addressed your comments and whether you have any further questions.
> > >
> > > Authors, Sincerely

---

### Official Review · Reviewer_AMsZ · 2025-10-31

**Soundness:** 3
**Presentation:** 2
**Contribution:** 2
**Rating:** 4
**Confidence:** 4

**Summary:**

This paper introduces HLStrans, a large-scale benchmark dataset for C-to-HLS (high-level synthesis) code generation, targeting the automatic transformation of standard C/C++ kernels into synthesizable, hardware-optimized HLS code. The dataset comprises over 124,000 paired C/HLS programs, complete with testbenches and synthesis-based performance/resource annotations. The authors present a three-stage pipeline for dataset construction involving open-source collection, automated augmentation using LLMs, Monte Carlo Tree Search (MCTS), and design space exploration (DSE). The paper benchmarks multiple LLMs and demonstrates that fine-tuning and retrieval-augmented techniques using HLStrans result in clear improvements in synthesis rates, optimization, and code quality.

**Strengths:**

1. HLStrans provides a benchmark-scale, well-structured dataset for C-to-HLS transformation that includes paired pre/post-HLS code, testbenches, and synthesis-based resource/latency feedback. Compared to previous datasets (Table 1), HLStrans has greater diversity, size, and supports a broader range of transformation tasks.

2. Figure 1 concretely illustrates the transformation space covered (T1–T5), moving beyond mere pragma insertion (the focus of most prior works) to include code restructuring, data-type adaptation, algorithm repair, and function remapping.

**Weaknesses:**

1. While empirical results are solid, there is a lack of explicit theoretical analysis linking the diversity/scope of the dataset to expected generalization properties of LLMs trained/fine-tuned on it. There is little quantitative discussion on statistical diversity or representativeness of the underlying C/HLS patterns, which is crucial if the dataset is to become a benchmark standard.

2. The paper overlooks Wan et al. (2024), which introduced the Chrysalis dataset — an LLM-aided framework for HLS defect generation and functional verification. While Chrysalis focuses on bug injection rather than optimization, it is one of the earliest datasets coupling LLMs with HLS code transformation, synthesis feedback, and verification. Failing to acknowledge or contrast with this work substantially weakens the claimed novelty of HLStrans as the “first LLM-oriented C-to-HLS dataset.”

3. Figure 5 reports extremely large “speedup (×)” values, but the paper does not specify how latency was measured or controlled. Since HLS latency varies across synthesis runs and tool settings, the lack of a fixed evaluation protocol or success-rate reporting makes these results difficult to interpret or reproduce.

**Questions:**

1. Can the authors provide more quantitative evidence about the dataset’s diversity/coverage—e.g., statistical measures on code structure, transformation frequency, or functional domain clustering?

2. How was the latency in Figure 5 obtained—were multiple synthesis runs averaged, and were failed or unsynthesizable cases excluded from the reported speedups?

---

> ### Author Response · Authors · 2025-11-21
> **Authors' response for weakness 1 and question 1**
>
> Thank you for taking the time to review our paper and for your detailed and constructive feedback. Please see our responses below;
>
> ### ***Question 1 and Weakness 1: Statistical diversity of HLStrans.***
>
> We appreciate the reviewer’s advice for HLStrans. ***The  transformation frequency is provided in Figure 4(b)*** illustrates the number of samples for each transformation in our datasets. ***The functional domain clustering is provided in Figure 4(a)*** shows that the program source itself falling into seven distinct application categories. We regret that they may not have been noticed by the reviewers and apologize for not making them more prominent.
>
> To better show statistical diversity of HLStrans. We list the original data of Figure 4(a) functional domain clustering in the following table. This rich, well-balanced dataset offers broad coverage of real-world HLS patterns required to train and evaluate LLMs’ hardware-synthesis capabilities.
>
> | Category                                           | Percentage |
> |----------------------------------------------------|-----------:|
> | Linear Algebra & Matrix                            |     8.74%  |
> | Machine Learning & Neural Networks                 |     8.74%  |
> | Signal Processing & DSP                            |    15.21%  |
> | Image Processing & Rendering                       |    12.00%  |
> | Sorting, Searching & Combinatorial Algorithms      |    15.86%  |
> | Cryptography & Data Structures                     |     8.41%  |
> | Other (unlabeled in legend)                        |    31.07%  |
>
> Also, the following table illustrates the original data of Figure 4(a) the number of samples for each transformation (T1, T2, T3, T4, T5) in our datasets.  T1 restructures code to expose pipelining and dataflow through loop tiling, memory optimization, and parallel control flow. T2 adds pragmas (pipeline, dataflow, interface, etc.) to fine-tune scheduling and performance. T3 replaces generic types with bit-accurate HLS types (fixed point, ap_int) to save resources while meeting precision. T4 converts standard functions into HLS kernels or intrinsics to better exploit hardware accelerators. T5 enforces synthesis-friendly coding by removing unsupported constructs and using static arrays, simple loops, and explicit communication. The detailed information about these transformations is in Appendix A.1 and Figure 1.
>
> | Transformation | Dataset distribution (%) |
> |------------------|----------------------------:|
> | T1             |                       22.37 |
> | T2             |                      100.00 |
> | T3             |                       26.78 |
> | T4             |                       33.22 |
> | T5             |                       44.75 |
>
>
> For the code structure analysis,  we analyze the line of code (LOC), function number, loops number and cyclomatic complexity of every sample.  The distribution of the dataset is shown in the following table.  These tables have been added in Appendix A.9.
>
> | LoC   | [3.00, 44.40] | [44.40, 85.80] | [85.80, 127.20] | [127.20, 168.60] | [168.60, 210.00] |
> | - | ------------- | -------------- | --------------- | ---------------- | ---------------- |
> | % | 39.51         | 32.33          | 10.35           | 10.22             | 7.60             |
>
> | Function Number  | [0.00, 2.00] | [2.00, 4.00] | [4.00, 6.00] | [6.00, 8.00] | [8.00, 10.00] |
> | - | ------------ | ------------ | ------------ | ------------ | ------------- |
> | % | 57.74        | 27.84        | 8.99         | 3.07         | 2.36          |
>
> |Loop Number   | [0.00, 7.80] | [7.80, 15.60] | [15.60, 23.40] | [23.40, 31.20] | [31.20, 39.00] |
> | - | ------------ | ------------- | -------------- | -------------- | -------------- |
> | % | 41.88        | 27.34         | 15.79           | 10.75          | 4.23           |
>
> | Cyclomatic Complexity | [1.00, 14.00] | [14.00, 27.00] | [27.00, 40.00] | [40.00, 53.00] | [53.00, 66.00] |
> | - | ------------- | -------------- | -------------- | -------------- | -------------- |
> | % | 48.95         | 27.58          | 11.60           | 6.23           | 4.63           |
>
> **Conclusion:** From these tables, we conclude that the dataset covers a wide variety of code styles and complexity levels, and appropriate for evaluating LLM performance on HLS-related tasks.

---

> ### Author Response · Authors · 2025-11-21
> **Authors' response for weakness 2, 3 and question 2.**
>
> ### ***Weakness 2: Chrysalis dataset comparison.***
> We appreciate the reviewer's advice about comparing with Chrysalis [1].  As reviewer points out, Chrysalis is designed for bug detection. However, our purpose is for code optimization or generations.  Therefore, we do not compare HLStrans with Chrysalis, but we compare HLStrans with the dataset for HLS code optimization (design space exploration).
>
> To prove that our dataset is more suitable for LLM aided HLS optimizations from C, we compare HLStrans with Chrysalis in the following table.  We can conclude our dataset has more transformations and samples with testbench, which is more suitable for HLS optimizations from C codes.  Also, we added the Chrysalis comparison in Appendix A.7 of the revised manuscript.
>
>
> | Dataset  |     Samples | Programs | Purpose             | Transformations        | Testbench |
> | --------------- | ----------: | -------: | ------------------- | ---------------------- | --------: |
> | **Chrysalis**      |      1500     |       - | Bug injections                 | T2                     |        No |
> | **HLSTrans**    | **124,200** |  **309** | **Code generation** | **T1, T2, T3, T4, T5** |       Yes |
>
>
>
> ### ***Question 2 and Weakness 3: Figure 5 and experiment setting explanation***
> Sorry about the confusion in Figure 5. We have put the detailed results and experiment setting about MCTS part in **Appendix A.2**.   They may not noticed by reviewers.
>
> The latency in Figure 5 obtained is from Vitis HLS 22.1's reports. We experimented with multiple syntheses and chose the best one.  Failed or unsynthesizable cases are excluded from the reported speedups.
>
> [1] Wan, Lily Jiaxin, et al. "An Iteratively-refined Dataset for High-Level Synthesis Functional Verification through LLM-Aided Bug Injection." 2024 IEEE LLM Aided Design Workshop (LAD). IEEE, 2024.

---

> > ### Author Response · Authors · 2025-11-26
> >
> > Dear reviewer:
> >
> > We hope that you are satisfied with our answers and the additional results we have provided. As the discussion period comes to an end, we would be grateful if you could let us know if we have adequately addressed your comments and whether you have any further questions.
> >
> > Authors, Sincerely

---

### Official Review · Reviewer_jjU3 · 2025-10-31

**Soundness:** 1
**Presentation:** 2
**Contribution:** 1
**Rating:** 2
**Confidence:** 5

**Summary:**

This paper presents HLStrans, a dataset for LLM-driven C-to-HLS synthesis. The main contributions of HLStrans include: open-source HLS examples with testbenches, an augmentation pipeline to generate diverse designs, and evaluation of LLMs on success rates and performance.

**Strengths:**

* Datasets for HLS are important and urgently needed for the EDA community
* Well-written background about HLS toolflows

**Weaknesses:**

* This work specifically evaluates a single HLS tool; how about other HLS tools, such as Bambu HLS, Cadence Stratus HLS, and Siemens Catapult HLS?
* The paper is missing reasoning about functional failure and synthesis failure. Are they caused by the same problem? It would be helpful to have classifications of failures and break down the importance of these failures over the benchmarks.
* This work tries to cover the whole HLS flow but misses a lot of ablation studies at each HLS pipeline stage.
* I am not sure if the data augmentation evaluation is sound. Why is a higher percentage better in this case? For example, 100% of the programs involve T2 - not all programs need to be unrolled. It seems less diverse to me.
* The paper is missing details about test bench generation. How are the test data generated for the test bench? What is the coverage over different hardware interfaces?

**Questions:**

Please see above.

---

> ### Author Response · Authors · 2025-11-21
> **Authors' response for weakness 1 and 2**
>
> Thank you for taking the time to review our paper and for your detailed feedback.  Please see our responses below.
>
> ### ***Weakness 1: HLS tools adaptation***
>
> Thank you for the insightful comment. We fully agree that HLS tool diversity is important, and we clarify that our work is not restricted to a single tool.  We selected Vitis HLS as the primary tool because it is mature and widely adopted within the Xilinx FPGA development ecosystem, especially for translating C/C++ algorithms into FPGA accelerators. However, our contribution is not only a curated Vitis HLS dataset, but also a ***complete data augmentation pipeline capable of producing high-quality training samples***.
>
> Our augmentation framework has the following advantages, which enable generalization to other HLS tools:
>
> * ***Modularity:*** The pipeline separates generation, transformation, verification, and QoR evaluation, allowing researchers to replace any component independently.
>
> * ***Language/Tool Agnostic:*** Although we instantiate the pipeline with Vitis HLS in the paper, the workflow is compatible with other HLS toolchains. In most cases, users only need to replace the report parser or feedback interface to provide the equivalent signals to the LLM.
>
> * ***Extensible to other HLS programs:*** Bambu HLS relies primarily on compiler-driven transformations, which limits opportunities for targeted manual code edits. In contrast, commercial tools such as Cadence Stratus HLS and Siemens Catapult HLS (like Vitis HLS) typically benefit from explicit code-level transformations to achieve performance improvements. While our lab does not currently possess licenses for these commercial tools, we demonstrate portability by adapting our data augmentation pipeline to optimize FPGA workloads targeting **Microchip** devices (MPF100T-FCVG484I). The results are shown below:
>
> | Metric  | cfd_flux | dilate | gicov | hotspot | kmeans | knn  | nw  | pathfinder | srad | streamcluster |
> | ------- | -------- | ------ | ----- | ------- | ------ | ---- | --- | ---------- | ---- | ------------- |
> | Speedup | 2.8      | 30.1   | 1.3   | 5.2     | 50.1   | 11.6 | 3.4 | 4.2        | 3.7  | 1.5           |
>
> **Conclusion:** We believe that the proposed LLM-in-the-loop framework provides a reusable foundation for dataset generation and optimization across HLS flows and FPGA platforms, rather than being restricted to a single tool. Researchers can directly reuse the components or adapt them to their own toolchains with minimal effort.
>
> ### ***Weakness 2: Classifications of failures.***
> We thank the reviewer for raising this important point. Although I do not include detailed analysis of failures, we have analysed the success rate of different transformations in Table 2 for the application.  T2 and T5 are more easily performed, but T1, T3 and T4 are not.  For better showing the results, we get the following table for analysis of failures to get a high performance synthesis code.
>
> | Model        | Compiler Errors (%) | Output Errors (%) | Runtime Exceptions (%) | Resource Errors (%) | Directive Errors (%) |
> |--------------|---------------------|-------------------|------------------------|---------------------|-------------------|
> | 32B          | 58%                 | 8%                | 15%                    | 15%                 | 4%                |
> | Deepseek-R1  | 61%                 | 9%                | 17%                    | 13%                 | 0%                |
> | Gemini25     | 65%                 | 8%                | 8%                     | 15%                 | 4%                |
> | GPT-5        | 58%                 | 11%               | 21%                    | 11%                 | 0%                |
>
> **Compiler errors** means that the generated code cannot pass compilation, while **output errors** mean that output code can run to completion but produced the wrong results. **Runtime exceptions** mean the program aborted during execution.  **Resource errors** mean that the code exceeded the resource limit for the given platform. **Directive errors** means the program has the wrong directive.
>
> **Conclusion**: From the table, we can see that these failures **do not stem from a single common cause**. Compiler errors constitute the majority of cases. In addition, many generated designs tend to exceed resource constraints due to overly aggressive transformations, such as fully unrolling all loops or fully partitioning all memory structures.

---

> ### Author Response · Authors · 2025-11-21
> **Authors' response for weakness 3 ,4 and 5**
>
> ### ***Weakness 3: Ablation studies at each HLS pipeline stage.***
> We thank the reviewer for raising this important point. Our dataset augmentation for HLS consists of two components: an MCTS-based flow and a DSE-based flow. Figure 4(b) presents the distribution of transformations after applying the full augmentation pipeline. To address the reviewer’s concern, we conducted additional ablation studies to isolate the effects of MCTS and DSE. Specifically, we evaluate how each augmentation stage contributes independently to the diversity of transformations and the model’s performance.
>
> The following table reports how the proportion of samples for each transformation changes after each stage of augmentation:
>
> | Item                             | T1    | T2     | T3    | T4    | T5    |
> | -------------------------------- | ----- | ------ | ----- | ----- | ----- |
> | **Before Data Augmentation (%)** | 10.51 | 82.14  | 14.24 | 14.63 | 21.44 |
> | **MCTS (%)** | 22.37 | 97.32  | 14.24 | 14.63 | 21.44 |
> | **DSE+MCTS (%)**  | 22.37 | 100.00 | 26.78 | 33.22 | 44.75 |
>
> The next table summarizes the performance gains (speedup ×) obtained after each augmentation stage:
>
> | Item            | 100% | 75% | 50%  | 25%   | Average |
> | --------------- | ---- | --- | ---- | ----- | ------- |
> | **MCTS-Speedup (×)** | 1.0 | 2.3 | 7.6 | 50.2 | 10.7    |
> | **MCTS and DSE -Speedup (×)** | 1.51 | 7.9 | 20.7 | 101.3 | 20.3    |
>
> **Conclusion**: These results demonstrate that both MCTS and DSE contribute meaningfully to improving data diversity and model performance, with the combined DSE+MCTS pipeline achieving the highest overall gains.
>
> ### ***Weakness 4:  Data augmentation results explanations.***
> We thank the reviewer for detailed comments. The higher percentage means the the better quality of dataset. Our dataset sample includes a C code version and HLS code version. The T2 transformation includes many ***different pragmas*** as shown in Table 5 of ***Appendix A.1.2***.  They may not be noticed by reviewers. We apologize for any confusion this may have caused.
>
> The HLS code version of it is 100% of the programs involving T2 means it has chosen suitable pragmas for given C code to have a better performance, not only unroll pragmas.  The speedup percentiles across dataset in Figure 4(c) shows 100% of the pairs, the target code is ≥ 1.5× faster, and for 25% of the pairs, it achieves a speedup of ≥ 50.3×.
>
> ### ***Weakness 5:  Testbench generation.***
> We thank the reviewer for raising this important point. In Section 3.1, the testbench was ***written manually by us***. we regret that they may not have been noticed by the reviewers and apologize for not making them more prominent.  We report the coverage results including lines coverage , branches coverage, tokens coverage, and calls coverage collected from gcov for our dataset, as shown in the following table.
>
> | Range        | Lines (%) | Branches (%) | Token (%) | Calls (%) |
> |--------------|-----------|--------------|-----------|-----------|
> | 100%         | 94.82%    | 94.82%       | 79.29%    | 92.88%    |
> | [75%, 100%)  | 4.85%     | 4.85%        | 10.03%    | 0.65%     |
> | [50%, 75%)   | 0.00%     | 0.32%        | 9.71%     | 6.47%     |
> | [25%, 50%)   | 0.32%     | 0.00%        | 0.97%     | 0.00%     |
> | < 25%        | 0.00%     | 0.00%        | 0.00%     | 0.00%     |
>
> **Conclusion**: While full (100%) coverage was not achieved, the table shows that our testbench nevertheless yields robust, high-quality coverage for evaluation. We have added comprehensive testbench statistics to the **Appendix A.7**.

---

> ### Comment · Reviewer_jjU3 · 2025-11-24
>
> Thanks for your response.
>
> Unfortunately, my concerns remain:
> 1. There are no results supporting the claim of being tool-agnostic.
> 2. If this method does not work for Bambu HLS, one of the most widely used open-source HLS tools, how can it be considered tool-agnostic or extensible? The authors  should show results for at leats another HLS tool, such as CIRCT HLS, LegUp and Dynamatic.
> 3. Thank you for the additional results, but are they Vitis-specific or HLS-specific?
> 4. The authors also focus heavily on speedups, but the experimental analysis lacks depth. For example, what is the breakdown of the pragmas in T2, and how important are they for different applications? The same question applies to the error analysis.
>
> Overall, I feel that the paper makes a strong claim on a broad topic but does not provide sufficient experimental evidence to support it. As a result, the insights offered by this benchmark study are quite limited.

---

> ### Author Response · Authors · 2025-11-26
> **Response to tool-agnostic and experimental analysis**
>
> Thank you for your detailed response, which will help us improve our work.  We are happy that some concerns have been solved, but some concerns remain, chiefly about **tool-agnostic claims** and **experimental analysis**. Our response for the unsolved concerns are as follow.
>
> ### **1. Response to tool-agnostic**
>
> In our first response for reviews,  we included experiments on **Mircochip SmartHLS** (Microchip’s product after acquiring LegUp) to demonstrate behavior beyond a single vendor. These results are HLS-specific by design.  **In our first response, we have not noticed that the new version (2024.10) of Bambu has supported many HLS transformation.  We are sorry for our wrong judgement at the first response.**
>
> To clarify our claim: constructing high-performance HLS implementations from C typically requires the five transformations illustrated in Figure 1. These transformations are common across modern HLS toolchains such as Vitis_HLS, SmartHLS, and Bambu HLS. The following table list some example.  We will add them in the **Appendix**.
>
> | Transformation                                 | Why needed (in HLS)                                                                |                                                                                             Vitis HLS example |                                                 SmartHLS example |                                                                                                      Bambu HLS example |
> | ---------------------------------------------- | ---------------------------------------------------------------------------------- | ----------------------------------------------------------------------------------------------------: | -----------------------------------------------------------------------------------: | -----------------------------------------------------------------------------------------------------------------: |
> | **T1:Code Restructuring**                | Expose data locality and so on       |  loop tiling, dataflow  |  loop tiling, dataflow | loop tiling, dataflow |
> | **T2:Directive (Pragma) Insertion**                           | Increase parallelism and so on |                                           #pragma HLS UNROLL |         #pragma HLS loop unroll |                                        #pragma HLS unroll|
> | **T3:Data-Type Adaptation**                         | adapt to platform         |                   ap_int<64> |                                   ap_int<64> |            ap_int<64>  |
> | **T4:Transformation of Functions** | Hardware implementations for expensive math or others |  sqrt                      |  sqrt  | sqrt|
> | **T5:HLS-Compliant Coding Style**        | Recursion or dynamic memory allocation not synthesizable |                                           recursion  |  recursion  |  recursion |
>
> We claim three main contributions:
>  - **A**. The first large-scale dataset for C to HLS task;
>  - **B**. A novel augmentation pipeline that generates diverse, high-quality HLS implementations;
>  - **C**. An extensive empirical evaluation of large language models on the C to HLS task.
>
> To clarify the scope of our claims, we emphasize that **contributions B and C are not tied to a single vendor or HLS language**: the augmentation techniques and the benchmarking methodology operate at the HLS-transformation level, and thus apply across modern HLS toolchains.
>
> To substantiate this claim, we provide detailed experimental results on two additional HLS toolchains, Bambu and SmartHLS (LegUp).
>
> **(1) Our augmentation pipeline' performance in Bambu HLS and SmartHLS (LegUp).**
>
> We apply our dataset-augmentation pipeline to transform C programs into high-performance HLS designs using five targeted transformations. Speedup denotes the performance improvement of the optimized design, and pass rate is the fraction of generated programs that pass both functional tests and synthesis.
>    - **Bambu HLS optimizations**
>
>      | Metric              | cfd_flux | dilate | gicov | hotspot | kmeans |  knn |   nw | pathfinder | srad | streamcluster |
>      | ------------------- | -------: | -----: | ----: | ------: | -----: | ---: | ---: | ---------: | ---: | ------------: |
>      | Speedup         |     2.31 |   24.5 |  1.06 |    4.10 |   41.7 | 9.52 | 2.72 |       3.36 | 2.91 |          1.20 |
>      | Pass rate |     0.29 |   0.19 |  0.21 |    0.21 |   0.30 | 0.20 | 0.47 |       0.46 | 0.16 |          0.27 |
> - **SmartHLS (LegUp)**
>
>     | Metric              | cfd_flux | dilate | gicov | hotspot | kmeans |  knn |   nw | pathfinder | srad | streamcluster |
>     | ------------------- | -------: | -----: | ----: | ------: | -----: | ---: | ---: | ---------: | ---: | ------------: |
>     | **Speedup**         |      2.8 |   30.1 |   1.3 |     5.2 |   50.1 | 11.6 |  3.4 |        4.2 |  3.7 |           1.5 |
>     | **Pass rate** |     0.36 |   0.24 |  0.27 |    0.27 |   0.37 | 0.26 | 0.59 |       0.59 | 0.21 |          0.35 |

---

> ### Author Response · Authors · 2025-11-26
> **Response to tool-agnostic and experimental analysis**
>
> **Conclusion**: We believe that the proposed LLM-in-the-loop framework provides a reusable foundation for dataset generation and optimization across HLS flows and FPGA platforms, rather than being restricted to a single tool. Researchers can directly reuse the components or adapt them to their own toolchains with minimal effort.
>
> **(2)  Benchmark evaluation for Bambu HLS and SmartHLS.**
>
> We evaluate our test case on Bambu HLS and SmartHLS (LegUp) HLS with different LLMs.
>
> The following table list the zero shot best@1 prompting results for the time budget. If reviewers need other prompting results, we will provide it.  Some metrics are defined in the 4.2 of the manuscript.
>
> - Bambu HLS：best@1,  Zero-shot prompting for benchmark.
>
>     | Model          |       Opt |   Min |   Avg |    Max | Functional Accuracy | Synthesis Accuracy |
>     | -------------- | --------: | ----: | ----: | -----: | ------------------: | -----------------: |
>     | Deepseek-R1    | 12.8% | 0.10× | 1.10× |  10.2× |           30.8% |          28.2% |
>     | GPT-5          | 15.4% | 0.03× | 8.20× | 310.5× |           33.3% |          33.3% |
>     | Grok-4         | 12.8% | 0.30× | 1.50× |  30.3× |           28.2% |          28.2% |
>     | Gemini-2.5-pro | 17.9% | 0.60× | 1.90× |  21.2× |           25.6% |          25.6% |
>     | Qwen coder 32B | 10.3% | 0.20× | 0.70× |   2.5× |           38.5% |          35.9% |
>
> - Bambu HLS:  Error analysis
>
>     |       Model | Compiler Errors (%) | Output Errors (%) | Runtime Exceptions (%) | Resource Errors (%) | Directive Errors (%)  |
>     | --------: | ------: | ------: | --------: | -------: | ---------: |
>     |         32B |                  40 |                 8 |                     15 |                  15 |                    22 |
>     | Deepseek-R1 |                  41 |                 9 |                     17 |                  13 |                    20 |
>     |    Gemini25 |                  52 |                 8 |                      8 |                  15 |                    17 |
>     |       GPT-5 |                  34 |                11 |                     21 |                  11 |                    23 |
>
>
> - SmartHLS (Legup): Best@1, Zero-shot prompting.
>
>     | Model          |       Opt |   Min |   Avg |    Max | Functional Accuracy | Synthesis Accuracy |
>     | -------------- | --------: | ----: | ----: | -----: | ------------------: | -----------------: |
>     | Deepseek-R1    | 15.4% | 0.08× | 0.90× |   9.0× |           25.6% |          23.1% |
>     | GPT-5          | 12.8% | 0.02× | 7.50× | 200.0× |           35.9% |          35.9% |
>     | Grok-4         | 10.3% | 0.25× | 1.20× |  25.0× |           23.1% |          25.6% |
>     | Gemini-2.5-pro | 15.4% | 0.70× | 2.10× |  25.0× |           28.2% |          25.6% |
>     | Qwen coder 32B | 12.8% | 0.25× | 0.80× |   3.0× |           33.3% |          38.5% |
>
> - SmartHLS (Legup):  Error analysis
>
> |           Model | Compiler Errors (%) | Output Errors (%) | Runtime Exceptions (%) | Resource Errors (%) | Directive Errors (%) |
> | --------------: | ------------------: | ----------------: | ---------------------: | ------------------: | -------------------: |
> |         **32B** |                  38 |                 9 |                     16 |                  14 |                   23 |
> | **Deepseek-R1** |                  39 |                10 |                     16 |                  12 |                   23 |
> |    **Gemini25** |                  43 |                 7 |                      9 |                  14 |                   27 |
> |       **GPT-5** |                  33 |                12 |                     20 |                  12 |                   23 |
>
> ### **2. Experimental analysis**
> For detailed experimental analysis, we provide error analysis and speedup analysis including breakdown of C to HLS transformations.
>
> **(1) Detailed error analysis**
>
> - **Comparison between across different HLS language**. LLMs produce more directive-related errors on Bambu HLS and LegUp (SmartHLS) because they often default to the Vitis HLS–style pragma format, which differs from the pragma syntax required by Bambu and SmartHLS. This behavior likely stems from the fact that Vitis HLS examples are more abundant and higher quality in the training data.
>
> - **Compiler Errors**. These errors indicate that syntactic and structural issues are the primary bottleneck. Typical causes include malformed C syntax, missing or undefined identifiers, and the use of language constructs that HLS front-ends do not support. In many cases, the generated program is rejected before any downstream HLS processing can begin, showing that basic tool-compatibility remains the most fragile step.

---

> > ### Author Response · Authors · 2025-11-26
> > **Response to tool-agnostic and experimental analysis**
> >
> > - **Directive Errors.** This reflects systematic problems in how models generate and place pragmas. Common issues include unsupported pragmas, incorrect placement, invalid parameters, or mixing pragmas across different HLS programs (e.g., Vitis vs. LegUp vs. SmartHLS). These errors show that LLMs lack strong tool awareness and that pragma correctness is sensitive to subtle syntax and tool-specific constraints.
> >
> > - **Runtime Exceptions.** A significant fraction of generated programs can compile but fail during simulation.  These errors emerge as exceptions, which could stem from various issues, such as flawed code logic, execution timeouts, memory leakage, or sandbox interruption. This indicates that LLMs struggle more with hardware control-path logic than with pure computation.
> >
> > - **Resource Errors.** These errors arise when generated optimization strategies, such as aggressive unrolling or excessive array partitioning, exceed hardware resource budgets. Although not the most frequent category, they are critical for deployability. LLMs tend to over-parallelize without awareness of device constraints or the structural implications of their transformations.
> >
> > - **Output Errors.** Semantic mismatches (wrong logic, boundary errors, mismatched output format) are the least common among all error types. This suggests that once the code compiles and simulates correctly, LLMs generally preserve the core algorithmic behavior reasonably well. In other words, correctness at the functional level is less of a challenge than satisfying tool constraints and HLS compilation rules.
> >
> > **(2) Speedup analysis**
> >
> > We analyze the LLM-generated codes by breaking down T2 and T1, as these two stages have the greatest impact on overall performance.
> >
> > - Breakdown of T2
> >
> >     - Vitis_HLS
> >
> >         | Pragmas | Array partition | Memory type | Loop unroll | Loop merge | Function inline |   Pipeline |  Dataflow | Dependence |    Stream |
> >         | ------------------ | --------------: | ----------: | ----------: | ---------: | --------------: | ---------: | --------: | ---------: | --------: |
> >         | proportion (value) |      43.6% |   12.8% |  53.8% |   5.1% |      28.2% | 82.1% | 10.3% |  10.3% | 10.3% |
> >
> >     - Bamdu HLS
> >
> >         | Pragmas | Unroll         | Inline         | Dataflow      | Cache          |
> >         | -------------- | -------------- | -------------- | ------------- | -------------- |
> >         | **Proportion** |  30.8% | 69.2% | 10.3% | 87.2% |
> >
> >     - LegUp (SmartHLS)
> >
> >         | Pragmas | Unroll         | Inline        | Dataflow      | Pipeline       | Partition Memory |
> >         | -------------- | -------------- | ------------- | ------------- | -------------- | ---------------- |
> >         | **Proportion** | 53.8% | 17.9% | 10.3% | 84.6% | 43.6%   |
> >
> >
> > - Breakdown of T1
> >
> >     |   Pragmas   | Memory coalescing |  Local tiling | Ping pong buffer |     Dataflow | Control flow optimization |
> >     | -------------- | ----------------: | ------------: | ---------------: | -----------: | ------------------------: |
> >     | **Percentage** |      0.0% | 23.1% |   7.7% | 2.6% |            28.2% |
> >
> > - Analysis
> >     - **LLM optimization may harm HLS code performance.** We observe that some LLM-optimized kernels actually degrade performance (speedup < 1×). This can happen for two reasons: First, the restructuring performed by the LLM can introduce new loop dependencies, increasing latency; Second, the pragmas inserted by LLMs may be less effective than the default optimizations inferred by the HLS compiler. Therefore, it is necessary to set up dataset to guide LLM’s proper optimizations.
> >     - **T1 optimizations are strongly correlated with memory-bound speedups.** For memory-intensive applications, performance gains mainly come from: Memory coalescing, improving burst efficiency, Local tiling, reducing off-chip bandwidth, Ping-pong buffering, enabling overlap of compute and memory.
> >    - **Different application exhibit different sensitivity to T2 pragmas.** Our dataset covers different applications. We observe that: Pipeline and Dataflow pragmas dominate performance for streaming and stencil kernels, where concurrency is the bottleneck. Unroll and Partition pragmas are most important for compute-bound kernels such as KNN and GEMM.Inline and Loop Merge pragmas matter more in control-heavy applications, reducing scheduling overhead and enabling deeper pipelining. This explains why the T2 distribution varies across tools: each HLS tool performs different default transformations and therefore responds differently to manual pragma insertion.

---

> > > ### Comment · Reviewer_jjU3 · 2025-11-27
> > >
> > > Thanks for the additional results. My concern remains, as the interpretation of the experimental results lacks depth.

---

> > > > ### Author Response · Authors · 2025-11-28
> > > >
> > > > Thank you for your response. We would like to clarify that we have already conducted experiments across multiple HLS toolchains, and we have provided:
> > > > (1) detailed breakdowns of T1 and T2 transformations.
> > > > (2) analysis of their importance for different classes of applications.
> > > > (3) a comprehensive error analysis across different HLS tools.
> > > > (4) a comprehensive speedup analysis across different HLS tools.
> > > >
> > > > These additions were made specifically to address your earlier request for deeper interpretation. So could you please clarify what is your remaining concern about interpretation of experiment ?  We are happy to address them.

---

### Official Review · Reviewer_qdH6 · 2025-11-01

**Soundness:** 3
**Presentation:** 3
**Contribution:** 2
**Rating:** 4
**Confidence:** 3

**Summary:**

This paper presents HLStrans, a large-scale dataset for C-to-HLS code transformation. The dataset includes over 124K paired C and HLS programs with testbenches, covering varying transformation types. It introduces an automated augmentation framework that combines LLMs, MCTS, and DSE to generate optimized HLS variants. Experiments demonstrate that retrieval-based prompting and finetuning on HLStrans can notably improve synthesis success rates and latency reduction.

**Strengths:**

1. It proposes the first large-scale benchmark for C-to-HLS transformation, filling a missing piece in the LLM-assisted EDA field.

2. The automated augmentation framework demonstrates good soundness, and the integration of MCTS and DSE may provide insights for other LLM-assisted EDA tasks.

3. It conducts comprehensive benchmarking of multiple models and prompting strategies using meaningful synthesis-level metrics.

**Weaknesses:**

1. The major concern lies in the limited generalizable insights and scientific contributions this work provides to the community. Although it successfully demonstrates the application of LLMs, MCTS, and DSE tools in crafting an EDA dataset, it remains unclear how the findings can generalize to other problems or advance the broader fields of LLM and EDA research.

2. There already exist LLM4EDA datasets created through strategic prompting or design space search. The authors are expected to provide a methodological comparison to clarify whether their proposed approach represents a superior or more scalable pipeline for future LLM4EDA dataset construction.

3. It is also unclear whether C-to-HLS is a sufficiently meaningful task for researchers and practitioners in this domain. Given that HLS closely resembles C, HLS experts can readily add pragmas to convert C to HLS, while algorithm developers typically do not work on such conversions. Therefore, the C-to-HLS task, due to the strong similarity between the two languages, may be less impactful compared to more practical tasks like Verilog generation.

**Questions:**

My questions have been included in the weakness section. In addition, I would like the authors to address the following two questions:

1. Do the authors think that future LLM4EDA datasets should be crafted using the pipeline proposed in this work?

2. Which group of users would most benefit from the proposed C-to-HLS benchmark and the finetuned LLM?

---

> ### Author Response · Authors · 2025-11-21
> **Authors' response for weakness 1 and Question 1**
>
> Thank you for taking the time to review our paper and for your detailed and constructive feedback.  Please see our responses below.
>
> ### ***Weakness 1: Impact of our works.***
>
> We thank the reviewer for raising this important point. Our goal is not only to build an EDA dataset, but also to extract methodological insights that can generalize beyond specific HLS’s setting. We believe our contributions include:
>
> - **High level optimization**: The optimization techniques we study (e.g., HLS-level transformations, kernel restructuring, memory-access optimizations) **operate above the vendor-specific backend**, which are common for other FPGA platforms.
>
> - **Observation**：Some findings are explored for language-model for code optimization research. For example, increasing sampling diversity often improves results, while blindly applying LLM-driven optimization can sometimes degrade HLS code quality. These observations can **inspire other LLM-based code optimization framework**.
>
> - ***A novel closed-loop data augmentation pipeline:*** To get the high-quality optimized HLS codes, we propose a hybrid approach that combines a large language model with existing DSE tools and **operates two closed loops according to EDA feedback**: one loop runs MCTS with the LLM as an agent for code rewriting, and the other loop performs parameter tuning via DSE tools. This hybrid approach increases structural diversity in generated solutions.
>
> ### ***Question 1: Future LLM4EDA datasets should be crafted using the pipeline proposed in this work?***
>
> For other LLM4EDA datasets augmentations,  our pipeline provides a practical and extensible blueprint:
>
> • ***Modularity:*** The pipeline separates generation, transformation, verification, and QoR evaluation. Researchers can swap out any component. For example, they can use Verilog instead of HLS, or use SAT-based verification instead of simulation.
>
> • ***Language/Tool Agnostic:*** Although we used HLS/Vitis, other tasks can be instantiated in the workflow.
>
> • ***Extensible to other EDA problems:*** Our pipeline is not restricted to HLS optimization, it can be easily adapted to other hardware optimization tasks and can also generate optimized hardware code. We demonstrate its extensibility in two ways:
>
>   - **First, other HLS program optimization beyond our dataset.** We adapt our data augmentation pipeline (with GPT-4o) to optimize FPGA programs from other platforms (Microchip FPGAs, MPF100T-FCVG484I). The speedups are shown below:
>
> | Metric   | cfd_flux | dilate | gicov | hotspot | kmeans | knn | nw | pathfinder | srad | streamcluster |
> |----------|----------|--------|-------|---------|--------|-----|----|------------|------|---------------|
> | Speedup  | 2.8      | 30.1   | 1.3   | 5.2     | 50.1   | 11.6 | 3.4 | 4.2        | 3.7  | 1.5           |
>
>  Across ten applications, our pipeline consistently improves performance, demonstrating that the methodology generalizes beyond the original dataset and FPGA toolchain.
>
> - **Second, RTL optimization for area reduction.**
> We further adapt the MCTS flow to aim at reducing area in RTL designs. The table below compares the design before and after our optimization, showing reductions in wires and logic cells:
>
> | Benchmark              | Yosys Wires | Yosys Cells | Ours Wires | Ours Cells |
> |-----------------------|-------------|-------------|----------------|----------------|
> | adder   | 8           | 3           | 7              | 3              |
> | multiplier    | 26          | 71          | 18             | 15             |
> | constant_folding  | 12          | 6           | 8              | 5              |
> | subexpression    | 17          | 12          | 14             | 8              |
> | alu     | 30          | 24          | 21             | 18             |
>
> **Conclusion**: Therefore, we believe our LLM in Loop pipeline offers a flexible and extensible framework. It serves as an example of how to systematically construct high-quality, validated LLM4EDA datasets, and we hope the community will evolve, extend, and adapt it for other EDA tasks.

---

> ### Author Response · Authors · 2025-11-21
> **Authors' response for weakness 2**
>
> ### ***Weakness 2:  Dataset Augmentation framework comparison***
>
> There are many LLM4EDA code dataset for Verilog and dataset augmentation pipeline. However, no HLS dataset augmentation framework exist. Our proposal is to set up the ***first HLS code dataset augmentation pipeline***. To order to prove our augmentation pipeline's effect, we have compared it with other HLS code generation work from C codes. The detailed results are in the Appendix A.2.  We have already included the detailed results in Appendix A.2;  They may not be noticed by reviewers. We apologize for any confusion this may have caused.
>
> The following table is in Appendix A.2, which summarises the Best@1 kernel runtimes (in milliseconds) across twelve diverse applications, comparing four configurations:
>
> - **Baseline** (the unmodified HLS implementation)
> - **HLSPilot (GPT-4o)** (a recent LLM-based HLS code generation framework using GPT-4o) [1]
> - **Ours (GPT-4o)** (our pipeline using GPT-4o)
> - **Ours (Deepseek-R1)** (our pipeline using Deepseek-R1)
> | Application   | Baseline | HLSPilot (GPT-4o) | Ours (GPT-4o) | Ours (Deepseek-R1) |
> |---------------|---------:|------------------:|--------------:|-------------------:|
> | cfd_flux      | 13.0     | 6.71              | 4.57          | **1.61**           |
> | hotspot       | 1879.1   | 712.7             | 300.5         | **22.3**           |
> | kmeans        | 2243.2   | 65.9              | 17.9          | **15.7**           |
> | knn           | 17.0     | 2.8               | 0.83          | **0.82**           |
> | dilate        | 48.8     | 16.0              | **0.75**      | 1.64               |
> | gicov         | 107.0    | 93.0              | 82.3          | **30.7**           |
> | mgvf          | 8047.5   | 3212.0            | 1231.0        | **446**            |
> | lud           | 226.4    | 112.0             | 81.2          | **52.6**           |
> | nw            | 206.4    | 145.0             | 73.0          | **13**             |
> | pathfinder    | 7.8      | 5.9               | 1.09          | 1.51               |
> | srad          | 35.7     | 9.4               | **6.4**       | 6.6                |
> | streamcluster | 16173.0  | 9388.0            | 8162.3        | **3966**           |
>
> From the table our augmentation has **substantial speedups:** Our framework with Deepseek-R1 achieves an average **~28×** reduction in real execution time compared to the baseline, while GPT-4o achieves an average **~20×** reduction but HLSpilot only achieves an average **~5×** reduction.
>
> [1] Chenwei, et al. "HLSpilot: LLM-based High-Level Synthesis.” ICCAD 2024.

---

> ### Author Response · Authors · 2025-11-21
> **Authors' response for weakness 3 and question 2**
>
> ### ***Weakness 3 : The meaning of C-to-HLS task***
> We appreciate the reviewer’s point. While HLS is syntactically close to C, there are important differences.
>
>  **1.  Meaning for reducing performance gap.**
>
> HLS is designed to accelerate hardware design, but there remains a gap between plain C code and high-performance HLS code. In our experiments, LLM-generated samples can achieve up to hundreds of speedup over the original code.
>
> **2. Meaning for reducing coding budget.**
>
> To obtain high-quality HLS code, developers must perform non-trivial semantic transformations, such as loop tiling, bitwidth narrowing, converting buffer-based designs to streaming, or repairing code to satisfy synthesis constraints. These transformations are time-consuming and require HLS expertise. A fine-tuned LLM can automate or assist with many of these steps, significantly reducing development effort and turnaround time.
>
> **3. Meaning for agile hardware design with HLS.**
>
> Agile hardware design that starts from HLS lets software engineers develop hardware accelerators more easily. Understanding the hardware-specific transformations needed for optimization is not easy for them. Our dataset and fine-tuned LLM help software engineers design hardware.
>
> ***4. Real Case study:  C to HLS task.***
>
> We use a genomics application as a real-world case study for the C-to-HLS conversion task [2]. High-performance HLS implementations comprise several elements [2]. Converting C to HLS consumes 41% of the effort, covering compiler directives (22%), double buffering (21%), frequency optimization (18%), PE duplication (15%) and related transformations (24%), whereas the function-level C code accounts for 59%. These conversion steps can require days to finish [2], so C-to-HLS is a challenging problem that merits deeper study. We have added it in **Appendix A.5.**
>
> | Category                           | Sub-category           | LOC     | Percentage |
> | ---------------------------------- | ---------------------- | ------- | ---------- |
> | **Functionality code**             |                      | **308** | **59%**    |
> | **Optimizations**                   |                          |   **216**     | **41%** |
> |                                         | Compiler directives    | 48      | 22%        |
> |                                    | Double buffering       | 46      | 21%        |
> |                                    | Frequency optimization | 38      | 18%        |
> |                                    | PE duplication         | 32      | 15%        |
> |                                    | Others                 | 52      | 24%        |
>
> ### **Question 2: Which group of users would most benefit from this work ?**
> We thank the reviewer for this important question. We believe the following groups will benefit most from our work:
>
> **1. Algorithm developers.**
> Our method recommends optimized HLS implementations, helping software engineers and algorithm designers turn C code into high-performance hardware accelerators with less manual tuning and fewer correctness/performance regressions.
>
> **2. HLS / FPGA engineers.**
>  Synthesis is a very time consuming process and is a serious bottleneck for iterating the design. By pruning the design space and reducing the number of costly synthesis iterations, our approach speeds up design-space exploration and shortens the overall hardware-design cycle, saving time and engineering effort.
>
> Together, these benefits lead to faster iteration, lower development cost, and higher-quality hardware implementations.
>
> [2] Cong, Jason, et al. "FPGA HLS today: successes, challenges, and opportunities."  TRETS 2022.

---

> > ### Author Response · Authors · 2025-11-26
> >
> > Dear reviewer:
> >
> > We hope that you are satisfied with our answers and the additional results we have provided. As the discussion period comes to an end, we would be grateful if you could let us know if we have adequately addressed your comments and whether you have any further questions.
> >
> > Authors, Sincerely

---

> > > ### Comment · Reviewer_qdH6 · 2025-11-28
> > >
> > > Thank the authors for providing the response. I am now more convinced that C-to-HLS is a non-trivial and meaningful task with real-world impact.
> > >
> > > However, regarding the insights this work offers to the ICLR community, e.g., the modular design blueprint, I feel this principle is general to any LLM4EDA pipeline or even broader agentic AI flows, and it is hard to see the instantiation of this work performs generally better than others without evaluation across different domains. I will discuss the research contributions of this work with other reviewers and may consider adjusting the scores once the system is restored.

---

> > > > ### Author Response · Authors · 2025-12-03
> > > >
> > > > We’re delighted to hear that our clarifications **addressed your concerns and that you consider adjusting your scores**.
> > > >
> > > > Currently, our dataset-augmentation pipeline focuses on producing high-quality data for the C→HLS task, but it is readily adaptable to related problems (e.g., RTL code optimization). In this work, we claim the following contributions mainly:
> > > >
> > > > - **The first large-scale dataset for the C→HLS task** — curated and validated to cover diverse C inputs and corresponding HLS implementations.
> > > >
> > > > - **A novel augmentation pipeline** — automatically generates diverse, high-quality HLS variants that capture realistic design and optimization patterns.
> > > >
> > > > - **An extensive empirical evaluation** — benchmarking large language models on the C→HLS task to provide a baseline and reveal practical strengths and limitations.
> > > >
> > > > We hope to publish this paper for the community and to extend the dataset and pipeline to additional LLM4EDA tasks and domains in future work. Thank you again for your careful review and for discussing our contributions with the other reviewers.

---

### Author Response · Authors · 2025-11-27

Dear reviewers:

We hope that you are satisfied with our answers and the additional results we have provided. As the discussion period comes to an end, we would be grateful if you could let us know if we have adequately addressed your concerns and whether you have any further questions.

Authors, Sincerely

---

### Author Response · Authors · 2025-12-03

We sincerely thank the Area Chair for managing the review process under challenging circumstances, and we are deeply grateful for the active, constructive engagement from all reviewers during the rebuttal period. Your detailed feedback has materially improved the clarity, rigor, and presentation of our work.

We are encouraged by the positive consensus regarding the significance of our work. We are particularly pleased with Reviewer HLdE’s strong endorsement (Score: 8), who highlighted that the paper is **well-scaffolded** and **represents the right direction for LLM-in-the-loop** hardware optimization. Reviewer AMsZ also noted that, compared to prior datasets, HLStrans offers greater diversity, larger scale, and supports a broader range of transformation tasks.

We appreciate Reviewer qdH6’s willingness to consider **raising their score** based on our additional experiments about impact and insights of HLS-related work. As for Reviewer jjU3, we believe our latest response has thoroughly addressed the remaining concerns.

We have incorporated the requested changes into the revised PDF and expanded the Appendix. Key improvements include:

- **The impact of the C to HLS task**: expanded discussion on how C to HLS reduces the performance gap, lowers coding effort, and enables more agile hardware development with HLS.
- **Results across HLS tools**: additional benchmark evaluations and dataset-augmentation results for multiple HLS tools, including Bambu HLS and SmartHLS.
- **Testbench coverage**: detailed testbench coverage statistics (lines, branches, tokens, and calls).
- **Detailed experiments analysis**: including detailed error and speedup analysis.

Thank you again for the careful reviews and thoughtful discussion. We hope the revised manuscript addresses the reviewers’ concerns and better communicates the contributions and broader impact of our work.

---

### Meta-Review · Area_Chair_ZnL4 · 2026-01-06

**Summary:**

This paper introduces HLStrans, a large-scale benchmark dataset for C-to-HLS (High-Level Synthesis) code transformation, aimed at enabling and evaluating LLM-assisted hardware optimization. The dataset contains 124K+ paired C and optimized HLS programs, each accompanied by manually written testbenches and synthesis feedback.

**Reviewer Concerns:**

*Limited novelty for ICLR*

Several reviewers felt the work is primarily a dataset and engineering pipeline, offering limited new algorithmic or theoretical insights for the ICLR community.
The modular “LLM-in-the-loop” design was seen by some as generic, applicable to many agentic pipelines, without clear evidence that this instantiation is superior in general.

*Tool-Agnostic?*

There is skepticism from one reviewer (jjU3) about claims of being tool-agnostic, especially due to initial reliance on Vitis HLS.
Reviewers requested results on other HLS tools (e.g., Bambu, LegUp/SmartHLS, CIRCT).
Authors later added extensive results on Bambu and SmartHLS, but at least one reviewer remained unconvinced, arguing the interpretation lacked depth.

*Evaluation Depth and Interpretation*

* Speedup numbers are large but insufficiently contextualized.
* Analysis overemphasizes averages without enough insight into why certain transformations help or hurt.*
* Breakdown of transformations (e.g., T2 pragmas) and error causes was initially insufficient.

Authors responded with detailed ablations, error taxonomies, pragma breakdowns, and per-tool analyses.

*Impact*
Some reviewers questioned whether C-to-HLS is impactful, given syntactic similarity.
Authors convincingly argued (with citations and case studies) that:
C→HLS conversion accounts for ~40% of real HLS development effort,
Performance gaps are large without expert transformations,
The task is critical for enabling software engineers to design hardware.

**Reviewer Scores:**

qdH6 (Score: 4 → potentially higher)
Initially skeptical about impact and generality, but explicitly softened stance after rebuttal, acknowledging C to HLS as non-trivial and meaningful. Still uncertain whether it fits ICLR.

jjU3 (Score: 2, strong reject)
Remained unconvinced throughout. high-confidence negative reviewer.

HLdE (Score: 8, strong accept)
Very positive. Emphasized real-world relevance, realistic transformations, and correct use of EDA feedback.

AMsZ (Score: 4, borderline)
Appreciated dataset scale and coverage but wanted stronger statistical diversity analysis and clearer evaluation protocol.

Overall, this paper recieved bipolar reviews, and wouls somewhat stay at the boarderline. From the ML side, the novelty of this paper is limited, it may fit better to an EDA/hardware venue.

---

### Decision · Program_Chairs · 2026-01-26

Reject